# Unveiling Language Skills under Faithful Circuits

## Abstract

[Circuit decomposition and counterfactual-based pruning have become the cornerstone framework for mechanism interpretability. However, the unfaithfulness to the output due to cumulative bias in the pruning process hinders more complex and detailed mechanism exploration. To address this, we propose a novel circuit discovery framework that faithfully identifies circuit graphs. This framework contains three steps: firstly, the language model is decomposed into a fully linear graph consisting of disentangled "memory circuits"; secondly, greedy search is adopted to prune while ensuring output faithfulness; finally, we adopt causal analysis on the pruned circuit graph to identify salient circuit graph, estimated by counterfactuals and interventions. Our framework facilitates the discovery of complete circuit graphs and dissection of more complex mechanisms. To demonstrate this, we explored three generic language skills (*Previous Token Skill*, *Induction Skill* and *In-Context Learning Skill*). Using the circuit graphs discovered through our framework, we identify the complete *skill paths* of these skills.] Our experiments on various datasets confirm the correspondence between our identified skill paths and language skills, and validate three longstanding hypotheses: 1) Language skills are identifiable through circuit dissection; 2) Simple language skills reside in shallow layers, whereas complex language skills are found in deeper layers; 3) Complex language skills are formed on top of simpler language skills. Our codes are available at: `https://anonymous.4open.science/r/language_skill`.

## 1 Introduction

[Mechanism interpretability (Elhage et al., 2021; Conmy et al., 2023) is becoming crucial for understanding how language models work. A common approach (Conmy et al., 2023; Yao et al., 2024; Syed et al., 2023; Bhaskar et al., 2024) involves breaking down the model into disentangled, more linear components organized as a computational graph. By applying counterfactual techniques and pruning, less important connections are removed, leaving behind a smaller "circuit graph" that highlights the key components contributing to the model's output.]

[However, existing circuit discovery methods often fail to faithfully represent the output of the model. Specifically, substituting the model's forward process with a circuit graph does not ensure that the predicted output token remains consistent with the original output of the language model. This lack of faithfulness indicates that other yet-undiscovered circuits may significantly influence the output, undermining the argument that the circuit graph fully captures the underlying mechanisms. The core issue lies in the pruning strategies employed by these methods, which are typically optimized for counterfactual scenarios. Decisions to remove an edge are based on the changes in logits between the original output and a 'corrupted output.' As a result, the cumulative effect of removing many edges introduces biases that can ultimately alter the model's output.]

[To address this challenge, we propose a two-stage discovery process, decoupling faithful pruning and causal discovery. In the first stage, we employ a greedy search algorithm to identify non-contributing edges in the original circuit graph, under the condition that the original output remains the same after performing each pruning step. This stage ensures a faithful pruning result, keeping the outputs unchanged. The second stage identifies salient circuit graph using counterfactual and intervention techniques. Additionally, to achieve more precise discovery, we completely dissect the transformer model into fully disentangled and linear components, known as "*memory circuits*", with the addition

of "*compensation circuits*" to account for the non-linearity of the MLP module within the transformer, which has not been accomplished in previous works. In summary, our framework encompasses three steps: complete linear circuit decomposition, faithful pruning, and causal analysis.]

[Compared to existing methods, our approach has distinct advantages: the lossless and linear decomposition holds the potential to identify all components responding to a pattern, while faithful pruning and causal analysis enable us to dissect more complex patterns. To show the potential ability for discovering new insights, We select three generic and progressively complex skills which have been introduced in (cro, 2024; Ren et al., 2024; Edelman et al., 2024; Olsson et al., 2022): a) *Previous Token skill* which is responsible for receiving information from the previous token; b) *Induction Skill* which duplicates tokens with the same prefix; and c) *ICL Skill* which perform inference based on similar patterns appeared in demonstrations. Utilizing the circuit graph obtained from our three-step framework, we unveil the complete *skill paths* of these skills. These skill paths have confirmed some conjectures that have long remained unverified:]

1. **Identifiability**: Language skills are identifiable through circuit dissection and correspond to different circuit paths.

2. **Stratification**: Simple language skills reside in shallow layers, whereas complex language skills are found in deeper layers.

3. **Inclusiveness**: Complex language skills are formed on top of simpler language skills. For example, the Induction skill, dealing with text formatted as "*A B ... A*" and producing "*B*" at the end, requires the Previous Token skill to carry information from "*A*" to "*B*". The ICL Skill likewise consists of the Induction Skill as an essential mechanism.

In summary, our contributions are 3-fold:

- We propose a complete and faithful circuit discovery framework, providing a theoretical basis for addressing the research gap in mechanism interpretability.
- We devise a 3-step framework to extract the paths of generic language skills in language models.
- Our analysis and experiments verify three properties among the Previous Token Skill, Induction Skill, and ICL Skill, which include identifiability, stratification and inclusiveness.

## 2  A COMPARISON WITH RELATED WORK

[Existing methods have proposed various pruning strategies, including greedy search, such as ACDC (Conmy et al., 2023; Yao et al., 2024), attribute patching, such as EAP (Syed et al., 2023), and optimization search, such as Opt-Prun (Bhaskar et al., 2024). However, these pruning strategies cannot guarantee to reproduce the original output of the model, and hence not faithful in their discoveries.]

[Table 1 shows the comparison between the results obtained using the output of the pruned graph and the original LLM output results on several commonly used circuit datasets: IOI, greater than, and induction. (For specific experimental details, see Appendix A.) The circuit graphs obtained by these methods cannot fully recover the model's original output under lossless circuit decomposition. Theoretically, these pruning strategies decide whether to delete an edge by calculating its importance score, which is related to the change in the final logit. However, this does not guarantee that the logits of other candidates will not exceed the original output. Therefore, we adopt a more direct approach, conducting a greedy search under the condition that the top $n$ candidates remain unchanged.]

Table 1: Can existing pruning strategies really recover the original outputs? 'linear mlp' represents whether their components can decouple the influence on the MLP into a linear combination, and 'recover rate of original output (%)' represents the percentage of the pruning output that is the same as the original model output under our lossless decomposition.

| method | recover rate of original output (%) | | |
|---|---|---|---|
| | IOI | greater than | induction |
| ACDC | 56% | 31% | 67% |
| Opt-Prun | 59% | 29% | 62% |
| EAP | 41% | 22% | 55% |
| Ours | 100% | 100% | 100% |

# 3 METHOD

In this paper we propose a novel 3-step framework to extract the target language skills.

- **Step 1** (Section 3.1): We decouple the architecture of transformer language models into a combination of individual "Memory Circuits", which independently represents the minimum unit for reading memory. This results in a *Complete Circuit Graph*, $\mathcal{G}$.

- **Step 2** (Section 3.2): Keeping the destination token unchanged, we adopt greedy search to remove redundant edges in $\mathcal{G}$, retaining only those paths necessary for predicting the last (destination) token and resulting in an *Irreducible Circuit Graph*, $\mathcal{G}*$.

- **Step 3** (Section 3.3): We estimate the causal effect of each path in $\mathcal{G}*$ on the target skill and select those paths rendering most significant changes as the skill paths. The final graph formed by the skill paths is named as *Skill Circuit Graph*, denoted as $\mathcal{G}^S$.

## 3.1 MEMORY CIRCUIT

Building on the foundation of the Transformer Circuit (Elhage et al., 2021), we propose a complete decomposition of the transformer model including the MLP layers. Using tensor products ($\otimes$), we can represent any layer of the transformer model:

$$
\begin{aligned}
output &= (Id + Id \otimes W_{MLP}) \cdot (Id + \sum_{h \in H} A^h \otimes W_{OV}^h) \cdot X \\
&= (Id + \sum_{h \in H} A^h \otimes W_{OV}^h + Id \otimes W_{MLP} + \sum_{h \in H} A^h \otimes W_{MLP} W_{OV}^h) \cdot X
\end{aligned}
\tag{1}
$$

where $X$ represents the input representation in each layer and $H$ represents the number of attention heads. Matrix $A$ is given by the attention mechanism $A = softmax((XW_Q)(XW_K)^T)$, and $W_{MLP}$ involves the MLP operation with activation given by $atv(XW_{M1})W_{M2}$. $W_{OV} = W_O W_V$ refers to an "output-value" matrix which computes how each token affects the output if attended to, while $W_Q, W_K, W_V$ are parameter matrices for query, key and value. $W_{M1}$ and $W_{M2}$ are weight parameters in two linear layers. This equation simplifies both the attention and MLP modules into linear matrix mappings, describing how the paths from input to output for each layer are decoupled into four independent circuits: 1) $C^{self} = Id \cdot X$; 2) $C^{attn} = \sum_{h \in H} A^h \otimes W_{OV}^h \cdot X$; 3) $C^{mlp} = Id \otimes W_{MLP} \cdot X$; 4) $C^{attn+mlp} = \sum_{h \in H} A^h \otimes W_{MLP} W_{OV}^h \cdot X$. Moreover, three of these circuits can be further factorized as:

$$
C^{attn} = \sum_{h \in H} f_{W_{QK}}^{attn}(X) \cdot W_{OV}
\tag{2}
$$
$$
where \ f_{W_{QK}}^{attn}(X) = softmax((XW_Q)(XW_K)^T)X
$$
$$
C^{mlp} = f_{W_{M1}}^{mlp}(X) \cdot W_{M2}
\tag{3}
$$
$$
where \ f_{W_{M1}}^{mlp}(X) = atv(XW_{M1})
$$
$$
C^{attn+mlp} = \sum_{h \in H} f_{W_{QK}, W_{OV}, W_{M1}}^{attn+mlp}(X) \cdot W_{M2}
\tag{4}
$$
$$
where \ f_{W_{QK}, W_{OV}, W_{M1}}^{attn+mlp}(X) = atv(f_{W_{QK}}^{attn}(X)W_{OV}W_{M1})
$$

We use $f$ to represent a function that can be considered equivalent to an activation function, for instance, $f_{W_{QK}}^{attn}(X)$ represents the softmax-normalization of the input $X$ through a weighted accumulation performed by $QK$ values. In conclusion, these three types of circuits can be expressed using a common paradigm:

$$
C^{attn/mlp/attn+mlp} = f(X) \cdot W
\tag{5}
$$

The function $f(X)$ possesses the ability for non-linear transformations, while $W$ is an input-agnostic parameter, which can be understood as a memory learned through training (Geva et al., 2021). Therefore, this paradigm is capable of generating non-linear "weights" ($f(X)$) from the input representation $X$ and assigns these "weights" to a static memory distribution to extract the necessary "knowledge" for output. These three circuits thus represent the minimum and complete unit for

Table 2: Specific circuit index and corresponding implementation in each layer of GPT2-small. $W$ and $b$ represent weight and bias parameters, $atv$ represents the activation of MLP. $ln(\cdot)$ is the layernorm function. $A = softmax(XW_QW_K^TX^T + b_QW_K^TX^T + XW_Qb_K^T + b_Qb_K^T)$. Memory Circuits are $C^{1-25}$.

| Index | Category | Implementation($X$=input representation in each layer) |
|---|---|---|
| $C^0$ | Self | $X$ |
| $C^{1-12}$ | Attention | $A^h ln(X)W_V W_O + A^h b_V W_O$ |
| $C^{13}$ | MLP | $atv(ln(X)W_{M1})W_{M2}$ |
| $C^{14-25}$ | Attention+MLP | $atv(ln(A^h ln(X)W_V W_O + A^h b_V W_O)W_{M1})W_{M2}$ |
| $C^{26}$ | Compensation | $(atv(ln((\sum_{h=1}^{12} C^h)W_{M1})) - \sum_{h=1}^{12} atv(ln(C^h)W_{M1}))W_{M2}$ |
| $C^{27}$ | Compensation | $(atv((ln(C^{0-13})W_{M1}) - atv(ln(C^0)W_{M1}) - atv(ln(\sum_{h=1}^{12} C^h)W_{M1}))W_{M2}$ |
| $C^{28}$ | Bias | $b_v + atv(b_{M1})W_{M2} + b_{M2} + \sum_{h=1}^{12} act(b_V W_{M1})W_{M2}$ |

manipulating how much memory to read (i.e., memory-reading operation), and are independent of each other, which we refer to as **"Memory Circuits"**[1].

In this paper, we select GPT2-small as the target language model, containing 12 layers ($L = 12$) and 12 attention heads ($H = 12$). To provide a complete dissection of the the model at each layer which can precisely recover the original output, we introduce *Bias Circuits* and *Compensation Circuits* (Compensation circuits represent the synergy of the sum of linear terms passing the non-linear function, please refer to Appendix C for more details), apart from *Memory Circuits*, to compensate for the remaining information not covered by the memory circuits. Table 2 shows the specific circuits and their implementation for each layer. Our circuit dissection leads to a lossless decomposition of the original LM layer[2]: $LM_l(X) = \sum_{i=0}^{28} C^i$.

We treat Memory Circuits as the smallest units and build a Complete Circuit Graph, $\mathcal{G} = \{\mathcal{C}, \mathcal{E}\}$, where $\mathcal{C}$ stands for the set of 29 circuits ($C^{0-28}$ shown in Table 2, where Attention and Attention+MLP has 12 circuits due to 12 heads given) and $\mathcal{E}$ represents the path between any two circuits in different layers. Any memory circuit $C^i (0 \leqslant i \leqslant 25)$ in any layer $l (0 \leqslant l \leqslant 11)$, denoted as $C^{l,i}$, would receive information streams from all circuits in previous layers, i.e., $\mathcal{E} = \{(C^{l_1,i} \to C^{l_2,j})\}(0 \leqslant l_1 < l_2 \leqslant 11, 0 \leqslant i, j \leqslant 25)$. Notably, the lossless decomposition ensures that the insights gained from our circuit network accurately reflect the behavior of the original language model.

## 3.2 GREEDY SEARCH

Given the input tokens for LMs, $X = \{x_1, \cdots, x_{N-1}\}$, the whole optimization loss is:

$$\mathcal{L} = -\sum_{n=1}^{N} \log P(x_{n+1}|x_1, \cdots, x_n) \tag{6}$$

Without loss of generality and to facilitate our analysis, we focus on predicting the last destination token, $x_N$, given the historical context, i.e., $\mathcal{L}^{dst} = -\log(x_N|x_1, \cdots, x_{N-1})$. It can be reasonably hypothesized that many circuits and paths are not dedicated to the prediction of the destination token $x_N$ but related to other source tokens. Therefore, we need to prune the circuit graph and retain those paths that are essential for the prediction of destination tokens. This will afford a more explicit and causal view of the efforts made by the language model to generate $x_N$.

Specifically, we use a greedy search strategy to prune unnecessary paths between Memory Circuits while ensuring that the top $n$[3] candidates for the prediction of the destination token remain unchanged. Given that a depth-first search is more likely to remove shallow paths, we employ a breadth-first search

---

[1]Please note that while there are finer-grained functions in practice, such as $A \otimes X$, although filled with activation and attention, they suffer deep constraints to generate new vocabulary distribution and do not fully encompass the complete function. We elaborate in detail in Appendix B.

[2][In fact, due to the pytorch's floating-point calculation, there is an ignorable loss (minimum squared error between the sum of circuits and the original layer output $LM_l(X)$ is $< 10^{-11}$).]

[3]We set $n = 1$ in our experiments because our research model, GPT2-small does not consider candidates below top1 as outputs.

(We compared different search strategies and constraints in Appendix D) as shown in Algorithm 1: We

---

**Algorithm 1** Greedy Search for $\mathcal{G}*$

---

**Require**: Complete Circuit Graph $\mathcal{G} = \{\mathcal{C}, \mathcal{E}\}$, prediction $x_N = Model(\mathcal{G}, X)$, number of Layers $L$ and Circuit Index $[0, 28]$. **Ensure**: Irreducible Circuit Graph $\mathcal{G}* = \{\mathcal{C}, \mathcal{E}*\}$

   $\mathcal{G}* = \mathcal{G}, \mathcal{G}' = \mathcal{G}*$
   **for** each Memory Circuit $C^{l,i} \in \mathcal{C}(0 \leqslant l < L, 1 \leqslant i \leqslant 25)$ **do**
      **for** each Memory Circuit $\tilde{C}^{l',i'} \in \mathcal{C}(0 \leqslant l' < l, 1 \leqslant i' \leqslant 25)$ **do**
         $P = [[l', i', ], [l, i]], \mathcal{G}' = \mathcal{G}*, \mathcal{E}' = \mathcal{E} * -P$
         **if** $Model(\mathcal{G}', X) == x_N$ **then**
            $\mathcal{G}* = \mathcal{G}'$
         **else**
            $\mathcal{G}' = \mathcal{G}*$
         **end if**
      **end for**
   **end for**
   **return** $\mathcal{G}*$

---

denote $\mathcal{G}*$ as the Irreducible Circuit Graph after pruning, and $\mathcal{E}*$ as a subset of $\mathcal{E}$ which only includes those paths encapsulating the information stream necessary for the destination token prediction. $\mathcal{G}*$ thus represents the smallest, independent, and functionally complete circuit graph which is necessary for generating $x_N$.

### 3.3 ESTIMATION OF CAUSAL EFFECTS FOR LANGUAGE SKILLS

It is widely recognized that most texts require more than one language skill for inference (Arora & Goyal, 2023). Therefore, determining which paths are associated with the observed skill can be challenging. For this reason and motivated by endeavors in causal effect analysis (Wang et al., 2023; Vig et al., 2020), we divide the effects of any text on the output token into 3 components: **skill effects**, **background effects**, and **self effects for destination** (abbreviated as **self effects**).

Skill effects refer to the impact of the observed language skill on the output which is the focus of this paper. Self effects denote the impact of using a single destination token to predict, which functions like a "bi-gram model" (a model associating one input token with its output token). Background effects propose a counterfactual scenario, i.e., what would the effect be if this skill is not present in this text[4]. We use the typical example of the "Induction" skill for illustration, which works with an input in the form of "... A B ... A", where *A, B* refers to different tokens. Here the language model is expected to repeat the pattern ("*A B*") it has seen in the context and predict token "*B*" as the destination token.

Figure 1 illustrates an example of the "Induction" skill where the model outputs "*question*" when given the input "*Generate a question with a*". However, the vocabulary distribution in the output given by the language model does not merely result from the induction skill, but is also confounded by other effects such as the background effect and the self effect. To compute the target effect for a specific circuit path, let $Path^i$ be any directed paths in $\mathcal{G}*$ (e.g., $C^{1,19} \rightarrow C^{2,14} \rightarrow C^{6,5}$ *s.t.* circuit edges $(C^{1,19}, C^{2,14})$ and $(C^{2,14}, C^{6,5})$ are in $\mathcal{G}*$). $Path^i$ then symbolizes the flow of information across layers amongst the circuits it encompasses. We use the occurrence rate of $Path^i$ in all samples to

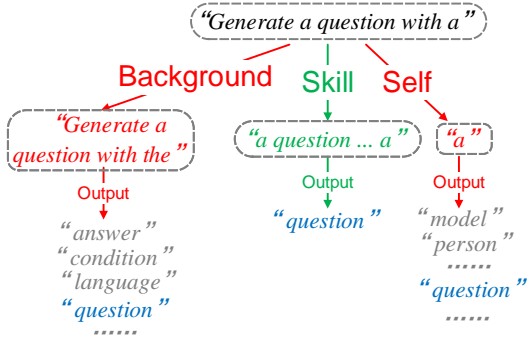

Figure 1: A case text about causal effects.

---

[4]Recognizing the impracticality of realizing the strict counterfactual scenarios, we adopt texts that are as close as possible to the input text, but without the observed skill, as counterfactual texts.

compute the effect:

$$Eff(Path^i_{\mathcal{G}*}) = \frac{N^{Path^i_{\mathcal{G}*}}_{Path^i_{\mathcal{G}*}=1}}{N_{all}} \tag{7}$$

$N^{Path^i_{\mathcal{G}*}}_{Path^i_{\mathcal{G}*}=1}$ represents the number of samples encompassing $Path^i$ while $N_{all}$ represents the number of all samples. Each path contributes differently to the three effects. Hence, we aim to find those paths that contribute to the skill effect rather than the other two effects.

Specifically, for each input text as a sample $s$, we perturb it to create a background text $s_{Bkg}$ and a self text $s_{Slf}$ (The process for generating background text and self text for all types of skills is described in Appendix E). Eventually, any sample is augmented with two more perturbed versions, rendering three types of inputs (i.e., original text, background text, and self text), each of which is subjected to the greedy search as discussed in Section 3.2. The greedy search produces three distinct Irreducible Circuit Graphs: $\mathcal{G}_{Ori}*$ (from original input text), $\mathcal{G}_{Bkg}*$ (from background text), and $\mathcal{G}_{Slf}*$ (from self text). Therefore, the skill effect (e.g., *Induction Skill*) of $Path^i$ can be defined as:

$$Eff_{Skill}(Path^i) = \frac{N^{Path^i}_{Path^i_{\mathcal{G}_{Ori}*}=1, Path^i_{\mathcal{G}_{Bkg}*}=0, Path^i_{\mathcal{G}_{Self}*}=0}}{N_{all}} \tag{8}$$

Finally, we get the Skill Circuit Graph $\mathcal{G}^S = \{\mathcal{C}, \mathcal{E}^S\}$. With $\delta$ as the threshold parameter: $\mathcal{E}^S = \{Path^i | Eff_{Skill}(Path^i) > \delta\}$ (we provided detailed analysis about $\delta$ in Appendix E.5).

# 4 EXPERIMENTAL DESIGN

This paper focuses on 3 language skills, spanning from basic to advanced levels:

**Previous Token Skill**: This is a skill to receive information from the previous token.

**Induction Skill**: This skill involves identifying patterns in prefix matching and replicating recurring token sequences.

**ICL Skill**: This is a complex skill to recognize and replicate the demonstration context, thereby producing outputs based on similar patterns.

Extensive research has shown that these three skills build on one another in a sequentially encompassing manner (cro, 2024; Olsson et al., 2022; Ren et al., 2024; Edelman et al., 2024). The Induction Skill inherently includes the Previous Token Skill. In simple terms, for induction to occur in the sequence "*A B ... A*", the token *B* must retrieve information from the preceding token *A*. Likewise, In-Context Learning must be capable of identifying similar patterns across different demonstrations to generate analogous outputs.

We select over 10k samples encompassing one of the three above-mentioned skills from large corpora and popular datasets such as WIKIQA (Yang et al., 2015), SST-2 (Socher et al., 2013), BIG-BENCH (Srivastava et al., 2023), OpenOrca (Lian et al., 2023), and OpenHermes (Teknium, 2023). For each instance, we create a background perturbation and a self perturbation (discussed in Section 3.3). For simplicity, **PVT** represents the sample set involving the Previous Token Skill and **IDT** represents the sample set related to Induction Skill. **ICL1** represents the ICL sample set from SST-2 datasets; **ICL2** represents the ICL sample set from object_counting task; **ICL3** and **ICL4** represents those from qawikidata and reasoning_about_colored_objects task. Using GPT2-small as the research model and applying the three-step framework detailed in Section 3 to these samples, we are able to identify high-effect samples through clustering, which clearly reveal distinct skill paths. The details of data preparation and implementation are elaborated in Appendix E, while our validation, findings, and explorations are presented in Sections 5, 6, and 7.

# 5 VALIDATION

## 5.1 WHEN SKILL PATHS ARE REMOVED

To understand whether the identified skill paths are responsible for their corresponding language skills, we design an intervention experiment by removing different sets of paths and observe the

Table 3: Accuracy of output to original label within different Circuit Graph

| Sample | Circuit Graph | | | | | | | | |
|---|---|---|---|---|---|---|---|---|---|
| | $\mathcal{G}*$ | $-R50$ | $-R500$ | $-\mathcal{G}^{S,PVT}$ | $-\mathcal{G}^{S,IDT}$ | $-\mathcal{G}^{S,ICL1}$ | $-\mathcal{G}^{S,ICL2}$ | $-\mathcal{G}^{S,ICL3}$ | $-\mathcal{G}^{S,ICL4}$ |
| PVT | 1.00 | 0.46 | 0.23 | 0.01 | 0.00 | 0.00 | 0.01 | 0.00 | 0.00 |
| IDT | 1.00 | 0.58 | 0.29 | 0.08 | 0.00 | 0.00 | 0.00 | 0.01 | 0.00 |
| ICL1 | 1.00 | 0.61 | 0.23 | 0.01 | 0.00 | 0.00 | 0.00 | 0.00 | 0.00 |
| ICL2 | 1.00 | 0.51 | 0.18 | 0.00 | 0.00 | 0.01 | 0.00 | 0.01 | 0.01 |
| ICL3 | 1.00 | 0.54 | 0.21 | 0.00 | 0.00 | 0.00 | 0.00 | 0.00 | 0.00 |
| ICL4 | 1.00 | 0.62 | 0.30 | 0.07 | 0.03 | 0.01 | 0.02 | 0.00 | 0.00 |

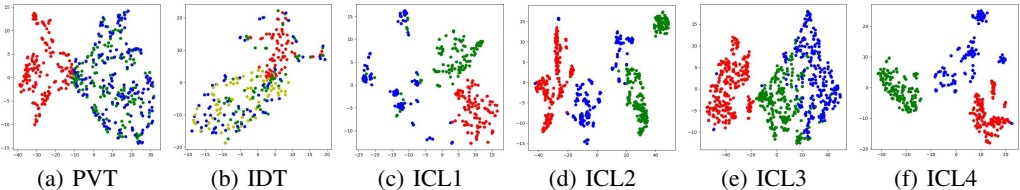

(a) PVT          (b) IDT          (c) ICL1          (d) ICL2          (e) ICL3          (f) ICL4

Figure 2: T-sne visualization of 6 types of samples on top 5 vocabulary candidates. Red denotes the original output model ($\mathcal{G}$), while blue signifies the output once a corresponding skill path is removed ($\mathcal{G} - \mathcal{G}^S$). The outputs for the background text ($\mathcal{G}_{Bkg}$) and self text ($\mathcal{G}_{Slf}$) are indicated in green and yellow, respectively.

output of the LM. Table 3 displays the accuracy of 6 types of samples under different configurations of the Circuit Graphs when treating the original output as the ground-truth. For each language skill $S$, we randomly select 500 samples from its corresponding dataset. As a result, 9 different configurations of Circuit Graphs are tested: $\mathcal{G}*$ which represents the original output; $-R50$ which signifies the removal of 50 paths at random from $\mathcal{G}*$; $-R500$ after the deletion of 500 paths randomly from $\mathcal{G}*$, which approximately equals the number of skill paths[5]. The remaining 6 configurations encompass the removal of paths from $\mathcal{G}*$ that correspond to the skill of Previous Token, Induction, ICL1, ICL2, ICL3, and ICL4, respectively (For additional supplementary data for this validation test, please refer to Appendix E.4.).

The results indicate that almost all samples were unable to produce the original token when these skill paths were excluded (as indicated in the last 6 columns), yet random removal of paths does not lead to such significant impact. Additionally, Figure 2 visualizes the t-SNE representation of the top 5 candidate outputs associated with different Circuit Graphs. It is clear that when a skill path is removed, the output (blue) shifts from red towards green (or yellow), indicating a transition from a text output distribution that includes skills to a distinct space resulted from the removal of these skills.

## 5.2 How Skill Effects Are Confounded

Another question is whether the background effect and self effect, mentioned in Section 3.3, potentially exist as confounders or share the circuits with observed skills. To answer this question, we conduct two experiments, with the results shown in Appendix F. Initially, Table 11 checks the overlap between the paths with $Eff > 0.5$ in the background/self text and the skill paths, illustrating that a small portion (approximately 10%-20%) of those paths does not belong to any observed skill. This corresponds to the confounding originating from other latent skills that we envisioned. Secondly, Figure 6 visualizes these different-effect paths' bivariate probability density function with the original input and background/self text. One intriguing discovery is that the confounding skills are more likely to present in the background text than in the self text, and the more complex the skill under analysis, the subtler the confounding effect introduced by the self text.

---

[5][The exact number of removed paths is: $-\mathcal{G}^{S,PVT}$ 325, $-\mathcal{G}^{S,IDT}$ 466, $-\mathcal{G}^{S,ICL1}$ 589, $-\mathcal{G}^{S,ICL2}$ 622, $-\mathcal{G}^{S,ICL3}$ 603, $-\mathcal{G}^{S,ICL4}$ 537]

Table 4: Key Receivers in Skill Circuit Graphs, green circuits are presented in the lower skill

| Skill | Receivers with receiving more than 10 paths ([#layer, #circuit]) |
|---|---|
| PVT | [1, 8], [1, 18], [1, 19], [1, 20], [1, 21], [2, 1], [2, 7], [2, 14], [2, 18], [2, 20], [2, 22], [2, 24], [11, 1], [11, 14] |
| IDT | [2, 14], [2, 18], [2, 20], [3, 14], [3, 17] [4, 5], [4, 12], [5, 11], [6, 5], [11, 1], [11, 14] |
| ICL1 | [2, 14], [2, 20], [2, 22], [2, 24], [3, 3], [3, 4], [3, 5], [3, 11], [3, 14], [3, 17], [4, 3], [4, 5], [5, 11], [8, 5], [10, 10], [11, 8], [11, 9], [11, 10], [11, 11] |
| ICL2 | [1, 19], [2, 14], [2, 20], [2, 24], [3, 5], [3, 11], [3, 14], [4, 5], [4, 7], [4, 9], [5, 10], [6, 5],[10, 9], [10, 10], [10, 11], [11, 1], [11, 5] |
| ICL3 | [1, 8], [1, 18], [1, 19], [1, 20], [1, 21], [2, 14], [2, 20], [2, 24], [3,1], [3, 14], [4, 3], [4, 5], [5, 1], [5, 10], [5, 11], [8, 1], [8, 9], [10, 5], [10, 10], [10, 12], [11, 1], [11, 8] |
| ICL4 | [1, 16], [1, 20], [2, 20], [4, 3], [4, 5], [5, 3], [6, 4], [6, 5], [8, 9], [9, 4], [9, 5], [10, 2], [10, 10], [10, 12], [11, 2], [11, 3], [11, 4], [11, 6], [11, 15] |

# 6 DISCOVERY OF LANGUAGE SKILLS

Table 4 displays the circuits receiving more than 10 circuit paths (receivers) in the skill graphs. We use $[l, i]$ to denote the circuit $C^{l,i}$ in the $l$-th layer and $i$-th circuit. The complete Skill Circuit Graph can be found in Appendix J. From Table 4, we identify 3 interesting patterns:

**1. Identifiability**: The paths of each skill are identifiable and remain unchanged across most data instances.

**2. Stratification**: The Previous Token Skill (PVT) is one of the simplest language skills, and thus it is located across layers 0-2. The Induction Skill (IDT) is slightly more complex and thus spreads across layers 0-6. Meanwhile, ICL is the most complex skill and has key receivers across nearly all layers. Additionally, all skills share the 11-th layer (final layer).

**3. Inclusiveness**: Higher-level skills always entail the key circuits of lower-level skills. It is universally acknowledged that the Previous Token Skill is an integral part of the Induction Skill, which is why circuits such as [2, 14], [2, 18] and [2, 20] (presented in PVT) can be found in the Induction Skill Graph. Similarly, the ICL skill encapsulates the Previous Token Skill and Induction Skill as necessary sub-skills, which is why circuits that are evident in the Previous Token Skill (such as [2, 14], [2, 20], [2, 24]) and those identified in the Induction Skill (such as [3, 14], [4, 5]) can be found in the ICL Skill Graph. Furthermore, we list all multi-step paths with inclusive sub-path in Appendix G.

Additionally, we have observed some differences in the receivers of different ICL tasks. Combined with the insights provided by Bayazit et al. (2023) and Bricken et al. (2023), we suspect that these differences arise from distinct circuits required to process domain-specific knowledge across different tasks. Based on the paths, attention weights, and cosine similarities of the representations (detailed results on attention weights can be found in Appendix H), we have identified several circuits with distinct characteristics (We demonstrate the performances of other circuit discovery methods in validating these conclusions in Appendix I.):

**Preceding Token Circuit**: Circuit [4, 12] performs a unique function, namely, when any token serves as a query token to attend other tokens, this circuit is shown to consistently carry significant information from its preceding token to the query token.

**Key Token Circuit**: Circuit [3, 14] exhibits a significantly different function from the others. This circuit consistently focuses on certain key tokens in the preceding text – such as the beginning, ending, and label prompts – and transmits this information to subsequent query tokens. Additionally, other key circuits in layers 3 and 4 partially undertake these functionalities.

**Opposite Circuit**: When using the last token of each input to produce the embedding for a specific circuit, we notice that the cosine similarity between Circuit [11, 14] and other key circuits is usually less than 0, especially with Circuit [11, 1], where the cosine similarity reaches to $-0.92$. Previous work (Wang et al., 2023) has mentioned this phenomenon, hypothesizing the reason to be controlling the variance of the loss function.

Table 5: Top 5 Receiver circuits appearing most frequently in skill paths presented in correct output samples but not incorrect samples.

| Type | Top-5 circuits with absence rate |
|------|----------------------------------|
| F_IDT | [2, 18] ($\downarrow$0.37), [2, 14] ($\downarrow$0.32), [11, 1] ($\downarrow$0.28), [2, 20] ($\downarrow$0.26), [2, 24] ($\downarrow$0.26) |
| F1_ICL | [2, 24] ($\downarrow$0.45), [2, 20] ($\downarrow$ 0.42), [2, 22] ($\downarrow$ 0.41), [1, 20] ($\downarrow$0.39), [2, 14] ($\downarrow$ 0.32) |
| F2_ICL | [3, 14] ($\downarrow$0.29), [4, 5] ($\downarrow$0.28), [10, 10] ($\downarrow$0.28), [8, 9]($\downarrow$0.24), [4, 12] ($\downarrow$0.22) |

# 7 EXPLORATION - WHY WRONG OUTPUTS?

In this section, we present a new direction for explaining and exploring common erroneous answers using Skill Circuit Graphs. Specifically, by contrasting the Skill Graphs of "incorrect" outputs with those of correct outputs, we can further diagnose what leads to the failure in skill execution. Table 5 illustrates the key circuits exhibiting the highest absent rate[6] between 3 "incorrect" and correct output types. Specifically, we investigate one erroneous type of output from an induction skill sample (F_IDT), and two types from ICL skill samples (F1_ICL, F2_ICL).

F_IDT refers to those samples wherein the input possesses an Induction pattern ("*A B ... A*"), but ultimately does not output *B*. F1_ICL denotes those samples wherein the output includes a word outside of the label options from the demonstrations, for example, a case where the input text "*[review1], label: positive, [review2], label: negative, [review3], label:*" unexpectedly produces "*the*". Such an error indicates that the language model did not capture the ICL template pattern in this case. F2_ICL involves samples that capture the template pattern yet still produce incorrect outputs, for example, cases where the correct output should be "*positive*", but the prediction is "*negative*". We compare the circuit graphs of these "incorrect" samples with the correct samples and identify the top 5 circuits with the highest absence rate.

Table 5 exhibits several interesting phenomena where the largest discrepancies between correct and incorrect samples in both F_IDT and F1_ICL occur on key circuits at layer 2. These circuits originate from the previous token skill, which handles the skill of receiving information from the previous token, such as the "A $\rightarrow$ B" in the induction template "*A B ... A*", as well as patterns such as "label $\rightarrow$ positive" in ICL. The loss of this skill—failure during the execution of the previous token skill—means that both the Induction skill and ICL skill cannot pass the duplicated prefix information to the next token, leading to template-based errors.

To further understand why these samples do not successfully execute the previous token skill, we perform a bi-clustering operation on the Previous Token Skill (experiment details are shown in Appendix E.2), yielding a cluster with $Eff < 0.2$ across most of all paths. We compared this cluster (termed the low-effect cluster) with another cluster (named high-effect cluster), with some samples as follows (All samples are from the original text of the Previous Token Skill, tokenized into two tokens):

**Low-effect cluster**:*"About to", " all these", " am a", " and win", " and select", " care over", "In Singapore", " in the", " is a", " it was", " than they", "The language", "The country", " the movie"*

**High-effect cluster**: *" 2002", "Adriano", "Ajinomoto", " becomes", "Could you", " don't", " ended up", "If the", " iPhone", " Knowledge", " stressful", "Windows", " Youtube's"*

It becomes obvious that in the context of an experimental setting lacking enough context, the previous token skill is performed only when there is a strong semantic relationship between the two tokens. For pairs of tokens where the semantic relation is not strong, there tends to be a reliance on the bi-gram model decision from the destination token.

Furthermore, for F2_ICL, the absence rate is relatively lower, suggesting that the source of the error might not be due to a single explicit cause. These circuits generally reside in the middle or even deeper layers, incorporating functions such as induction and summarization. However, to further

---

[6]Let $N_{C^{l,j}}^{+}$ and $N_{C^{l,j}}^{-}$ be the number of paths received by $C^{l,j}$ in correct and incorrect samples. The absence rate for each circuit is calculated as $(N_{C^{l,j}}^{+} - N_{C^{l,j}}^{-})/N_{C^{l,j}}^{+} \in [0, 1]$.

analyze this, we would need to delve into the representational level, which for the moment goes beyond the scope of this paper.

## 8   LIMITATION AND CONCLUSION

We have identified three pressing limitations that need to be addressed. The first is the **time complexity** of the greedy search the second is the lack of further examination on the **representational study**, and the third is **scalability**. Assuming the time for one inference of LLM as $O(1)$, the time complexity of a single greedy search would then be $O(L^2N^2)$, i.e., the square of the layer number times the number of circuits. If we can overlook this time-consuming process, then the $\mathcal{G}_*$ for each input would effectively facilitate training. In other words, $\mathcal{G}_*$ could directly instruct LLM which paths are essential and which are not, thus streamlining the training process. Despite the time complexity, we recall our contribution on the analysis of LMs which is usually more challenging and does not require large-scale inference. Additionally, the lack of research at the representational level hinders our progress in answering more complex questions such as why certain samples fail to trigger a skill. Recognized that this is a rather challenging topic, we leave it as a promising future work. Finally, we recognized the limitations of testing on a single model and specific skills. Although many studies have validated the GPT-2 series to have public trustworthiness for research in mechanistic interpretability, making us confident in its capacity to support our contribution—the pioneering work in discovering the theoretical foundation and experimental design of language skills—there remains ample scope for scalability across a variety of models and skills for future work.

In conclusion, we propose a novel framework including faithful pruning and linear decomposition to completely dissect the language model and discover key components leading to meaningful language skills. Our framework contains three steps, involving [decomposing the LM losslessly into circuits including memory, compensation, and bias circuits], pruning paths preserving the inference outcome, and identifying salient paths for language skills via causal analysis. Through this process, we are able to identify the skill paths necessary for a language model to process texts. Furthermore, we demonstrate several interesting findings validating existing hypotheses. For example, each language skill is bound to specific circuits, and more complex skills are associated with deeper circuits. Additionally, we find that the evolution of complex skills extends along the path of simpler skills they encompass, providing strong experimental support for research on emergence discoveries. Lastly, we explored attributions of error samples to the absence of certain skill circuits. These findings could potentially offer novel feedback for the training process. Overall, we believe that our thorough discovery of language skills can generate more insights into the exploration of language models.

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

## A    DETAILS ABOUT OUTPUT RECOVERY TESTS

We believe that although the graph is pruned, it should not change the next token output by the LLM. Therefore, we selected representative works from three pruning strategies and verified whether their outputs are the same as the original output of the language model on our lossless circuit decomposition. Specifically, we selected three datasets:

**IOIdataset** (Wang et al., 2023), which is used to discover the circuit for indirect object inference in the LLM.

**Greater than** (Hanna et al., 2024), which is used to discover the circuit for size comparison in the LLM.

**Induction** (Gokaslan & Cohen, 2019), which is used to discover the induction head and induction-related circuit in the LLM.

Then, we selected a representative work from each of the three different pruning strategies:

**ACDC** (Conmy et al., 2023), Automatic Circuit DisCovery, which calculates the importance score of each edge and performs a greedy search based on the score.

**Opt prun** (Bhaskar et al., 2024), which converts the importance score into an optimization function and assigns a learnable parameter to each edge to indicate whether an edge needs to be deleted.

**EAP** (Syed et al., 2023), or Edge Attribution Patching, which makes a linear approximation of activation patching to assign an importance score to each edge, and retains the top-$k$ edges.

The language model was chosen as GPT2-small. On each dataset, under our lossless circuit decomposition, i.e., memory, compensation, and bias circuit framework, we obtained the corresponding circuit graph according to the search strategy in the corresponding method paper with provided settings. For these circuit graphs, we obtained new outputs (considering only a token length) using their corresponding forward processes. We compared the new output tokens with the original output of GPT2-small. Table 1 shows the percentage of their similarity.

## B    ANALYSIS ABOUT MEMORY CIRCUITS

### B.1    WHY $A \otimes X$ IS NOT THE CIRCUIT WITH COMPLETE FUNCTION?

We use $X^{l,n}$ to denote the hidden state representation corresponding to the $n$-th token at the $l$-th layer, and $U$ represents the unembedding matrix. Therefore, for any representation $X^{l,n}$, we can obtain its vocabulary distribution, i.e., the logits for each token candidate, using $X^{l,n}U$. We adopt a sample text, *"Beats Music is owned by"*, as the input. Table 6 shows the logits corresponding to the words *" the"* and *" Apple"* when these tokens are converted to vocabulary embeddings.

Our expected correct output is such that after the last layer's representation is unembedded, the logits for *" Apple"* reach their peak. However, as shown in Table 6, after conducting an $A \otimes X$ operation on the 1st layer's representation, the logit range for *" Apple"* is $[80.49, 86.44]$, where $80.49$ corresponds to the attention weight of *" Music"* to *" by"* being 1, and $86.44$ represents the attention weight of *" Be"* to *" by"* being 1.

This situation exposes a significant drawback. In the representations of all previous tokens, the logits for *" the"* are always higher than those for *" Apple"*. Hence, no matter how many effects $A \otimes X$ operations performed, it remains impossible for the logits of *" Apple"* to surpass those of *" the"*. Therefore, although $A \otimes X$ incorporates an activation function such as $softmax$, it can only

Table 6: Logits of *"the"* and *"Apple"* when the representation in 1-st layer products unembedding matrix, with input *"Beats Music is owned by"*

| Logits | Tokens | | | | | |
|---|---|---|---|---|---|---|
| | *"Be"* | *"ats"* | *"Music"* | *"is"* | *"owned"* | *"by"* |
| *"the"* | 95.45 | 89.43 | 91.20 | 99.32 | 94.21 | 101.52 |
| *"Apple"* | 86.44 | 82.13 | 80.49 | 82.31 | 82.57 | 83.41 |

be considered as semi-activated (Elhage et al., 2021). We refer to this as a "deep constraint", that is, $A \otimes X$ cannot allow the representation of the destination token to exceed the upper and lower boundaries of the previous token's representation. This is why we assert that $A \otimes X$ lacks full functions, that is, it does not possess memory capability.

### B.2  How to explain Memory Circuits?

Let's likewise map all the Memory Circuits into the vocabulary space:

$$V = C \cdot U = f(X) \cdot W \cdot U = f(x) \cdot WU \tag{9}$$

Simply put, we assume $X \in \mathbb{R}^{N,D}$, $f(X) \in \mathbb{R}^{N,M}$, $W \in \mathbb{R}^{M,D}$, and $U \in \mathbb{R}^{D,E}$, where $N$ represents the number of tokens, $D$ denotes the dimensions in the residual stream, $M$ refers to the dimensions in the circuit (such as the dimensions in QKV or MLP), and $E$ signifies the length of the vocabulary list. Naturally, $WU \in \mathbb{R}^{M,E}$, which could be seen as a collection of $M$ vocabulary distributions. These vocabulary distributions are unaffected by the input tokens and thus can be considered as the acquired memory from training.

The function $f(X) \in \mathbb{R}^{N,M}$ acts like a weight which specifies how much each vocabulary distribution contributes to the output. This confirms why MLP is generally regarded as a memory storage, as its dimensions are usually significantly larger than those of QKV. Simultaneously, it also explains the advantage of MoE: providing a wider range of options for vocabulary distribution.

In the final analysis, the inference process of a language model can be seen as constituting 3 key components: **"memory"**, **"movement"**, and **"ensemble"**. **"Memory"** pertains to acquiring a new distribution from memory distribution, while **"movement"** involves transferring token information to subsequent tokens. Finally, **"ensemble"** refers to the process of combining representations from multiple circuits to produce the final representation. Within this process, Memory Circuits serve as the smallest units responsible for **"memory"** and also encompass independent operations of **"movement"** ($C^{1-12}$ and $C^{14-25}$). Furthermore, they form individual elements of the **"ensemble"**. Therefore, we examine the interrelationships (necessary paths) between Memory Circuits to understand the language skills of language models.

## C  Derivation of Compensation Circuits

The input of the MLP consists of two parts: the residual stream and the output of the attention. Due to the presence of nonlinear activation functions, the residual stream and attention are coupled in the input, making it impossible to isolate their impact on the MLP, thereby affecting the verification of pruning. To address this, we introduce a compensation circuit, decomposing the MLP into four parts:

$$atv((X + \sum_{h \in H} Attn^h)W_{M1})W_{M2} = (atv(XW_{M1}) + \sum_{h \in H} atv(Attn^h W_{M1]}))W_{M2} + Cps^1 + Cps^2$$

$$where\ Cps^1 = (atv((X + \sum_{h \in H} Attn^h)W_{M1}) - atv(XW_{M1}) - atv(\sum_{h \in H} Attn^h W_{M1}))W_{M2}$$

$$Cps^2 = (atv(\sum_{h \in H} Attn^h W_{M1}) - \sum_{h \in H} atv(Attn^h W_{M1}))W_{M2}$$

$$\tag{10}$$

where MLP operation with activation given by $atv((X + \sum_{h \in H} Attn^h)W_{M1})W_{M2}$ ($W_{M1}$ and $W_{M2}$ are weight parameters in two linear layers and $atv$ represents the activation function), $X$ represents the input representation in each layer and $H$ represents the number of attention heads, $Attn^h$ represents

Table 7: Logits of *"the"* and *"Apple"* when the representation in 1-st layer products unembedding matrix, with input *"Beats Music is owned by"*

| Metrics | Strategies | | | | | | | | | |
|---|---|---|---|---|---|---|---|---|---|---|
| | Breadth-1 | Breadth-2 | Breadth-3 | Breadth-4 | Depth | Top-2 | Top-5 | Top-10 | Loss-1 | Loss-2 |
| Deleted Path(%) | 69% | 68% | 68% | 70% | 9% | 32% | 14% | 2% | 25% | 34% |
| Hamming | 0 | 14 | 21 | 27 | 26457 | 12947 | 21639 | 44712 | 21773 | 16721 |

the output of $h$-th attention head, $Cps^1$ and $Cps^2$ are compensation circuit, representing the synergy effect of the residual stream $(X + \sum_{h \in H} Attn^h)$ and the sum of attention head $\sum_{h \in H} Attn^h$ respectively.

The compensation circuit calculates the synergy between the output when linear terms are summed before passing through a non-linear function, and the output passing through a non-linear function before summing. Therefore, the compensation circuit is dynamic and related to the input. From the perspective of the MLP, if we want the compensation circuit to be 0, then the input to the MLP must be reduced to only one or zero linear terms. This is an unlikely occurrence in practical pruning, so we assume that all edges of the compensation circuit always exist.

# D   SEARCH STRATEGIES

We conducted extensive comparisons w.r.t. two elements: breadth-first search and top1 candidate consistency. 1000 samples, each less than 30 tokens in length, were randomly selected from the WIKIQA dataset (Yang et al., 2015) and applied to different search strategies:

- Breadth-1: Breadth-first search was conducted on $C^{l,i}$ where $l$ varies from 0 to 11, and $i$ from 1 to 25.

- Breadth-2: The same breadth-first search was done on $C^{l,i}$, but with $l$ running from 0 to 11 and $i$ from 25 to 1.

- Breadth-3: $l$ spanned from 11 to 0 and $i$ from 25 to 1 while conducting breadth-first search on $C^{l,i}$.

- Breadth-4: The breadth-first search on $C^{l,i}$ was performed randomly.

- Depth: The depth-first search on $C^{l,i}$ was undertaken with $l$ ranging from 0 to 11 and $i$ from 1 to 25 (i.e., treating $C^{l,i}$ as the sender rather than the receiver).

- Top-2: Altered constraint to ensure top 2 candidates' token consistency.

- Top-5: Altered constraint to ensure top 5 candidates' token consistency.

- Top-10: Changed constraint to ensure top 10 candidates' token consistency.

- Loss-1: The constraint was modified to ensure that $x_N$'s loss does not exceed the original loss by more than 5.

- Loss-2: The constraint was changed to ensure the loss of $x_N$ does not exceed 100% of the original loss.

We measured two metrics: Deleted Path, which is the total number of deleted paths divided by the total number of paths and times 100%, and Hamming, which is the Hamming distance between $G*$ obtained from each strategy and $G*$ obtained from Breadth-1.

Table 7 presents the results of these methods. Notably, different search sequences of breadth-first search do not lead to significant discrepancies. Depth-first search methods, however, are not as effective as breadth-first searches in deleting a sufficient number of paths. Compared to the top 1 constraint, it is challenging for other constraints to delete an adequate quantity of paths. We posit that this is because GPT2-small is a simple model and does not possess the capability to randomly select candidates from the top N for output.

# E    DATA PREPARATION AND IMPLEMENTATIONS

## E.1    DATA PREPARATION

### E.1.1    PREVIOUS TOKEN SKILL

We randomly selected 40k text samples comprising two tokens - *"token0 token1"* - from the WIKIQA, OpenOrca, and OpenHermes corpora. In 20k of these samples, the two tokens made up one word, while in the remaining 20k, *"token0"* and *"token1"* belonged to two separate words. For the background text, we chose *"token0"*, and for the self text, we selected *" token1"*. A complete sample is as follows:

{*text: " that most", backgound_text: " that", self_text: " most", GPT2-small_output: " of"*}

### E.1.2    INDUCTION SKILL

The samples for the Induction Skill also come from WIKIQA, OpenOrca, and OpenHermes. We randomly selected 14k samples with the template *"... A1 B ... A2"*, where the destination token *" A2"* is the same as the preceding token *" A1"*, and the total token length of the sample does not exceed 30. For the background text, we removed *" A2"* and had GPT2-small produce a new but different token to replace *" A2"*, resulting in *"... A1 B ... C"*. Since *" C"* is semantically supplemented by the preceding text and differs from *" A2"*, it preserves semantics as much as possible without the Induction Skill. The self text is still token *" A2"*. A complete sample is as follows:

{*text: "chinese lesson 1.2: chinese", backgound_text: "chinese lesson 1.2: The", self_text: " chinese", GPT2-small_output: " lesson"*}

### E.1.3    ICL SKILL

The 4 types of ICL skill samples come from SST-2 dataset and the object_counting, qawikidata, reasoning_about_colored_objects datasets in BIGBENCH. These samples have been named by us as *icl_sst2, icl_oc, icl_qa, icl_raco*, with quantities of *1000, 284, 1000,* and *135* respectively. Each sample is required to contain two different labelled demonstrations and should be answerable correctly by GPT2-small. Here are examples of the four types of samples:

*icl_sst2*:

{*text: ", nor why he keeps being cast in action films when none of them are ever any good Sentiment: negative\nfunny , even punny 6 Sentiment: positive\nis that secret ballot is a comedy , both gentle and biting . Sentiment:", backgound_text: "is that secret ballot is a comedy , both gentle and biting . Sentiment:", self_text: " Sentiment:", GPT2-small_output: " positive"*}

*icl_oc*:

{*text: "I have a piano, a trombone, a violin, and a flute. How many musical instruments do I have?A: four\nI have a banana, a plum, a strawberry, a nectarine, an apple, a raspberry, an orange, a peach, a grape, and a blackberry. How many fruits do I have?A: ten\nI have a head of broccoli, a cauliflower, a stalk of celery, a cabbage, a potato, an onion, a yam, a garlic, a lettuce head, and a carrot. How many vegetables do I have?A:", backgound_text: "I have a head of broccoli, a cauliflower, a stalk of celery, a cabbage, a potato, an onion, a yam, a garlic, a lettuce head, and a carrot. How many vegetables do I have?A:", self_text: " A:", GPT2-small_output: " ten"*}

*icl_qa*:

{*text: "The country of University of Tsukuba is A: Japan\nThe sport played by Judit Polgár is A: chess\nThe country of citizenship of Théophile Gautier is A:", backgound_text: "The country of citizenship of Théophile Gautier is A:", self_text: " A:", GPT2-small_output: " France"*}

*icl_raco*:

{*text: "On the nightstand, you see the following objects arranged in a row: a black bracelet, a pink booklet, a blue cup, and a silver cat toy. What is the color of the object directly to the left of the pink object? A: black\nOn the floor, you see a bunch of objects arranged in a row: a red cup, a gold bracelet, a fuchsia puzzle, a purple stress ball, and a burgundy fidget spinner. What is the color of the*

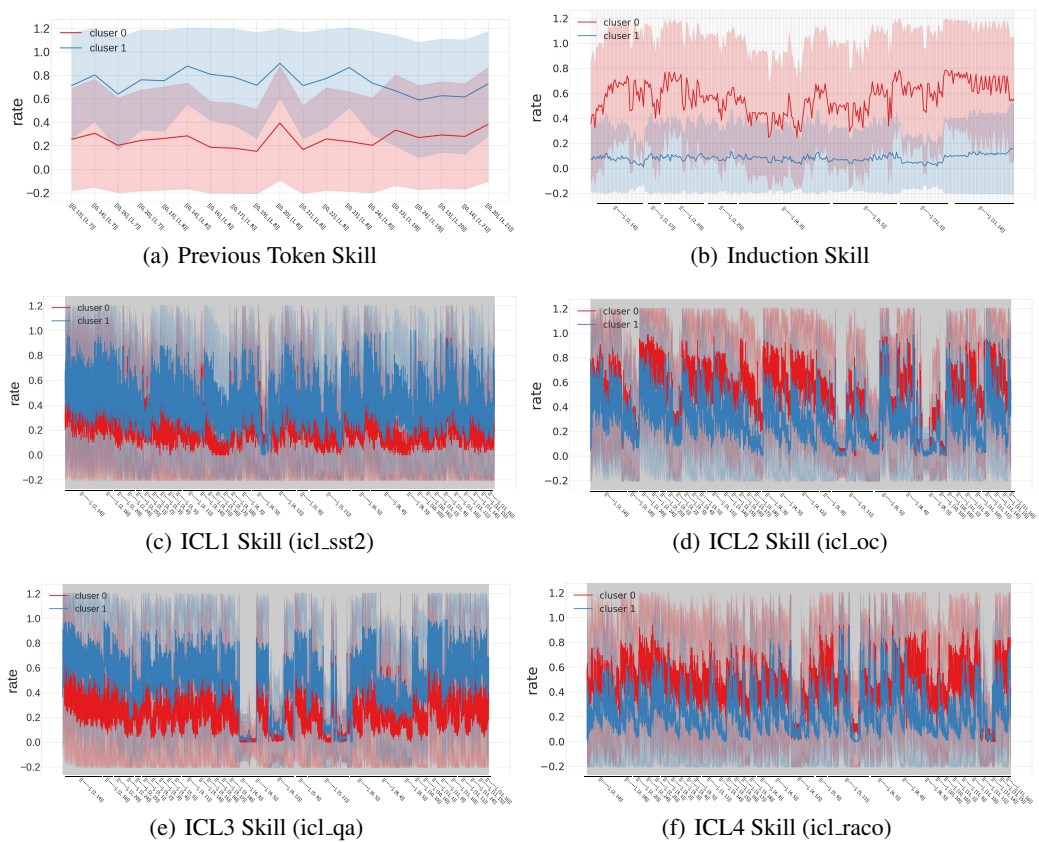

(a) Previous Token Skill

(b) Induction Skill

(c) ICL1 Skill (icl_sst2)

(d) ICL2 Skill (icl_oc)

(e) ICL3 Skill (icl_qa)

(f) ICL4 Skill (icl_raco)

Figure 3: bisection clustering on paths with top 10% $Eff_{Skill}$ for 3 skills

*object directly to the right of the cup? A: gold\nOn the table, you see a set of things arranged in a row: a black keychain, a purple mug, a blue dog leash, and a teal sheet of paper. What is the color of the left-most thing? A:", backgound_text: "On the table, you see a set of things arranged in a row: a black keychain, a purple mug, a blue dog leash, and a teal sheet of paper. What is the color of the left-most thing? A:", self_text: " A:", GPT2-small_output: " black"}*

### E.2 IMPLEMENTATION

In implementation, following the 3-step process from Section 3, we obtained the skill circuit graph, $\mathcal{G}^S$. We found that the skill effect values in $\mathcal{G}^S$ for the Previous Token Skill and the Induction Skill were not high, with the highest $Eff_{Skill}$ being only 0.54 and 0.61, respectively. However, the highest $Eff_{Skill}$ for the ICL Skill reached 0.98. We speculated that because the Previous Token Skill and the Induction Skill are overly simple, there were a significant number of samples that happened to output the correct answers without triggering the corresponding skill paths. For instance, in the text *"In China [mainland]"*, it's challenging to confidently determine whether *"mainland"* was influenced by the bi-gram model of *"China"* or if *"China"* received information from *"In"*. As such, we attempted to perform bisection clustering for each sample in the Previous Token Skill and Induction Skill, based on the paths with top 10% $Eff_{Skill}$.

Figure 3 shows the results of our clustering on the $\mathcal{G}^S$ for the 3 skills. The x-axis sequentially arranges the top 10% of paths on $Eff_{Skill}$ from shallow to deep, and the y-axis indicates the mean $Eff_{Skill}$ of these paths. It's striking that two clusters in the Previous Skill and Induction Skill: one consistently showing a high $Eff_{Skill}$, and the other showing little to no $Eff_{Skill}$. This suggests that these low $Eff_{Skill}$ samples hardly share common paths or trigger common language skills. Meanwhile, the ICL skill does not showcase discriminable clustering, further corroborating our speculation.

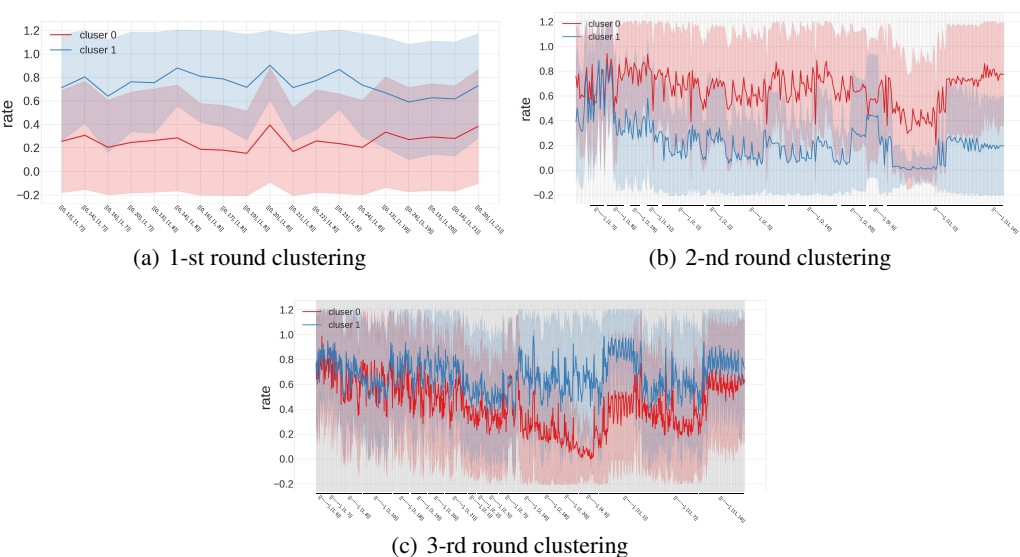

(a) 1-st round clustering    (b) 2-nd round clustering

(c) 3-rd round clustering

Figure 4: 3 rounds clustering in Previous Token Skill

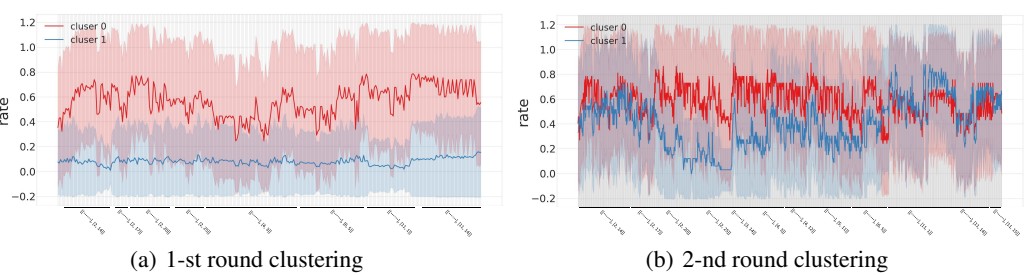

(a) 1-st round clustering    (b) 2-nd round clustering

Figure 5: 2 rounds clustering in Induction Skill

Going a step further, we would like to ascertain whether the Previous Token Skill and Induction Skill, after undergoing multiple rounds of "purification" through clustering, could still be divided into two clusters. Therefore, we recursively performed bisection clustering on the higher $Eff_{Skill}$ cluster each time. Figure 4 and 5 presents the results after each round of clustering. Notably, the Previous Token could not be divided after 2 rounds of clustering, while the Induction Token hit the dividing limit after just 1 round. Considering that the number of clustering rounds for ICL Skill was 0, we believe this supports our hypothesis: the more complicated the skill, the fewer instances of coincidental samples.

Lastly, we verified that bisection clustering significantly outperformed trisection, quad-section, and quintisection clustering. As illustrated in Figure 6, out of all the clusterings, only bisection clustering was able to distinctly segregate two mutually exclusive clusters categorized by high and low $Eff_{Skill}$.

## E.3 SENSITIVITY ABOUT BACKGROUND TEXT

To compare the sensitivity brought about by different background texts, we designed four different background text formats on the induction skill and compared the changes between the irreducible circuit graph ($G^*$) of these background texts and the final skill graph ($\mathcal{G}^S$). These formats are as follows:

**Bkg1**: For the induction text "......A1 B......A2", we replace *A2* with the output of the large model for "......A1 B......". For example, if the induction text is "*Chinese lesson 1.2: Chinese*", the background text is "*Chinese lesson 1.2: The*".

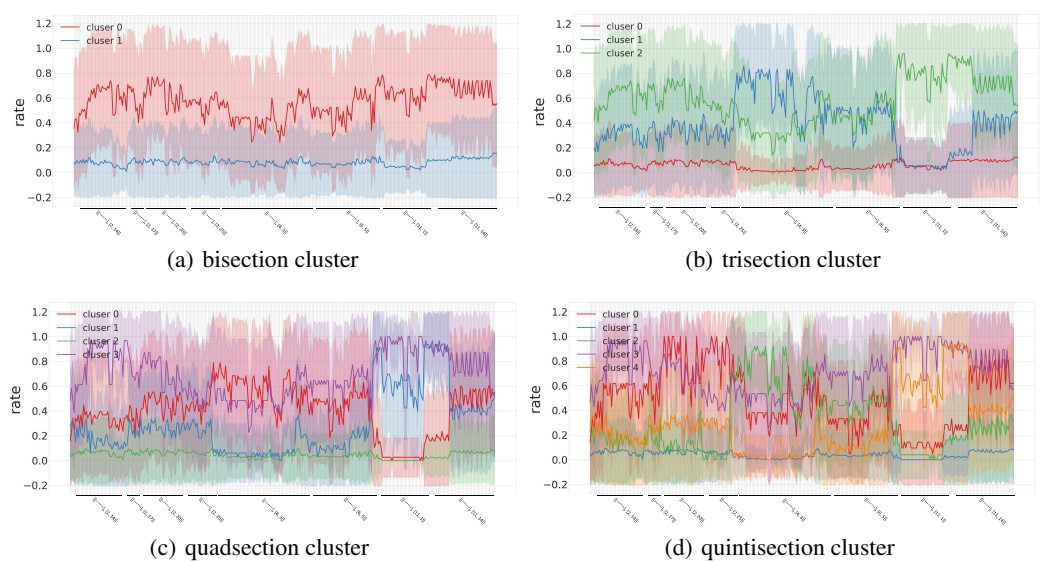

(a) bisection cluster       (b) trisection cluster

(c) quadsection cluster       (d) quintisection cluster

Figure 6: different clustering on Induction Skill

Table 8: HP between different background text. For example, the value in the second row and third column of Figure a is 6.42%, which means $HP(\mathcal{G}^*_{Bkg2}, \mathcal{G}^*_{Bkg3}) = 6.42\%$ ($\mathcal{G}^*_{Bkg2}$ and $\mathcal{G}^*_{Bkg3}$ has 6.42% edges different).

<table>
<tr><td colspan="5">(a) HP on $\mathcal{G}^*_{Bkg}$</td><td colspan="5">(b) HP on $\mathcal{G}^S$</td></tr>
<tr><td></td><td>Bkg1</td><td>Bkg2</td><td>Bkg3</td><td>Bkg4</td><td></td><td>Bkg1</td><td>Bkg2</td><td>Bkg3</td><td>Bkg4</td></tr>
<tr><td>Bkg1</td><td>0%</td><td>12.54%</td><td>9.33%</td><td>11.42%</td><td>Bkg1</td><td>0%</td><td>4.37%</td><td>5.75%</td><td>4.62%</td></tr>
<tr><td>Bkg2</td><td>12.54%</td><td>0%</td><td>6.42%</td><td>9.52%</td><td>Bkg2</td><td>4.37%</td><td>0%</td><td>3.51%</td><td>4.03%</td></tr>
<tr><td>Bkg3</td><td>9.33%</td><td>6.42%</td><td>0%</td><td>12.91%</td><td>Bkg3</td><td>5.75%</td><td>3.51%</td><td>0%</td><td>3.72%</td></tr>
<tr><td>Bkg4</td><td>11.42%</td><td>9.52%</td><td>12.91%</td><td>0%</td><td>Bkg4</td><td>4.62%</td><td>4.03%</td><td>3.72%</td><td>0%</td></tr>
</table>

**Bkg2**: For the induction text "*......A1 B......A2*", we directly delete *A2*. For example, if the induction text is "*Chinese lesson 1.2: Chinese*", the background text is "*Chinese lesson 1.2: *".

**Bkg3**: For the induction text "*......A1 B......A2*", we directly delete *A1*. For example, if the induction text is "*Chinese lesson 1.2: Chinese*", the background text is "* lesson 1.2: Chinese*".

**Bkg4**: For the induction text "*......A1 B......A2*", we replace *B* with the output of the large model for "*......A1*". For example, if the induction text is "*Chinese lesson 1.2: Chinese*", the background text is "*Chinese people 1.2: Chinese*".

To intuitively feel these changes, we introduced a metric of percentage Hamming distance, *HP*, specifically $HP(G_1, G_2) = hammingdistance(G_1, G_2)/(\sum_{G_1} \mathcal{E} + \sum_{G_2} \mathcal{E}) * 100\%$, i.e., when HP=0%, it means that the two graphs $G_1$ and $G_2$ completely overlap, and when HP=100%, it means that the two graphs do not overlap at all. We show the HP between $\mathcal{G}^*_{Bkg}$ and the HP between $\mathcal{G}^S$ under any two background texts in Tables 3 and 4.

### E.4 SUPPLEMENTARY DATA FOR VALIDATION

To enhance the transparency and validity of the validation experiment, we have supplemented it with some additional data.

Firstly, Table 3 only provides the accuracy of randomly deleting 50 and 500 edges, however, the dynamics of accuracy as the number of deleted edges changes is not disclosed. Therefore, we demonstrate the dynamics of accuracy in Figure 7 when the number of randomly deleted edges ranges from 50 to 1000. Notably, even with 1000 edges randomly deleted, the accuracy still remains above

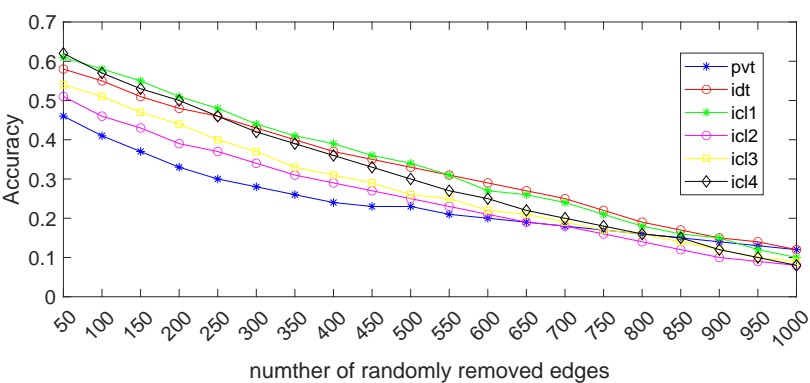

Figure 7: Accuracy with the number of removed edges increasing.

Table 9: Accuracy of output to original label within different Circuit Graph

| Sample Circuit Graph | $\mathcal{G}*$ | $-(\mathcal{G}^{S,PVT} - \mathcal{G}*)$ | $-(\mathcal{G}^{S,IDT} - \mathcal{G}*)$ | $-(\mathcal{G}^{S,ICL1} - \mathcal{G}*)$ | $-(\mathcal{G}^{S,ICL2} - \mathcal{G}*)$ | $-(\mathcal{G}^{S,ICL3} - \mathcal{G}*)$ | $-(\mathcal{G}^{S,ICL4} - \mathcal{G}*)$ |
|---|---|---|---|---|---|---|---|
| PVT | 1.00 | 1.00 | 0.88 | 0.89 | 0.89 | 0.83 | 0.89 |
| IDT | 1.00 | 0.93 | 1.00 | 0.81 | 0.82 | 0.85 | 0.81 |
| ICL1 | 1.00 | 0.95 | 0.81 | 1.00 | 0.95 | 0.93 | 0.97 |
| ICL2 | 1.00 | 0.93 | 0.84 | 1.00 | 0.92 | 0.95 | 0.92 |
| ICL3 | 1.00 | 0.94 | 0.86 | 1.00 | 0.93 | 0.91 | 0.94 |
| ICL4 | 1.00 | 0.96 | 0.83 | 1.00 | 0.93 | 0.94 | 0.96 |

0.1 (the total number of edges being considered is 6875). However, deleting the skill graph leads directly to an accuracy close to 0, even if the skill graph only contains around 500 edges. This further illustrates that the skill graph contains more edges that significantly determine the final output.

Secondly, in Table 3, we only showed the situation where low-level skill graphs remove those paths contained in high-level skill graphs. To reinforce the validation, we additionally provide in Table 9 the scenario where samples of low-level skills are only deleted from those edges that exist in the high-level skill graph but not in the low-level skills.

Herein, $-(\mathcal{G}^{S,PVT} - \mathcal{G}*)$ represents the deletion of paths in the previous token skill graph that do not exist in the target graph for the target sample, while $-(\mathcal{G}^{S,IDT} - \mathcal{G}*)$ represents the deletion of paths in the Induction skill graph that do not exist in the target graph. $-(\mathcal{G}^{S,ICL1} - \mathcal{G}*)$, $-(\mathcal{G}^{S,ICL2} - \mathcal{G}*)$, $-(\mathcal{G}^{S,ICL3} - \mathcal{G}*)$, and $-(\mathcal{G}^{S,ICL4} - \mathcal{G}*)$ respectively represent the deletion of paths in the ICL1, ICL2, ICL3, and ICL4 skill graphs that do not exist in the target graph for the target sample.

To reiterate, a portion of the paths in the high-level skill graph is identical to a portion of the paths in the low-level skill graph. Table 9 clearly demonstrates that when target samples delete those paths that exist in other skills but not in their own, the accuracy is not significantly affected. For instance, $-(\mathcal{G}^{S,IDT} - \mathcal{G}^{S,PVT})$ deletes 129 paths, but only reduces the sample accuracy of the previous token skill to 0.88, while the accuracy corresponding to randomly deleting 100 edges is only 0.42 (see Figure 7). In conjunction with Table 3, it explains that only the overlapping part of the Induction skill graph with the previous token skill graph affects the previous token skill. Additionally, when the ICL series skills output paths that exist in other ICLs but not in themselves, their accuracy is somewhat higher (over 0.9). This is due to the ICL series skill graphs being more similar to each other, resulting in fewer paths in the complement.

### E.5 THRESHOLD AND FAITHFULNESS

While we maintain faithfulness on $\mathcal{G}*$, it is difficult to maintain it on $\mathcal{G}^S$. In other words, the bias introduced by counterfactuals and interventions is indeed hard to completely avoid, while the faithfulness of pruning is avoidable. Therefore, a circuit graph that clearly reflects the final result will certainly discard some edges of unclear significance. This is usually accomplished through a threshold. We show in Figure 8 the change in accuracy when the threshold $\delta$ mentioned in Section 3.3 ranges from 0 to 0.9 (there are almost no circuits left when $\delta > 0.9$, so we ignore this part). It can

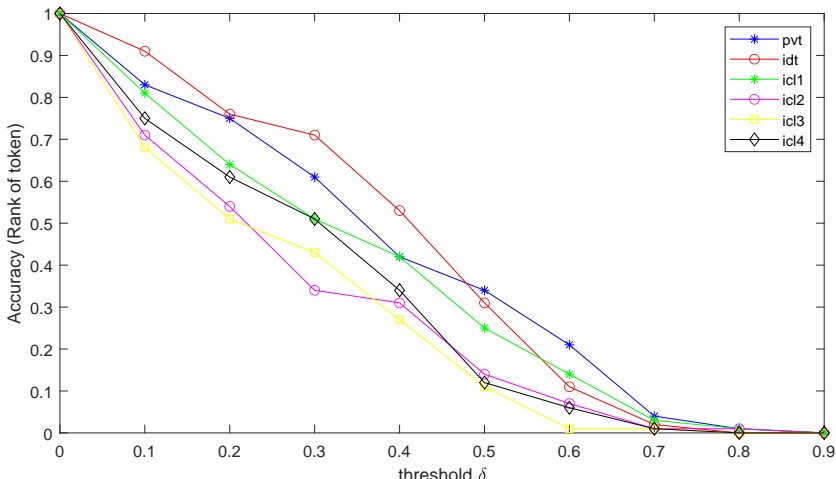

Figure 8: Faithfulness ranging from the $\delta$

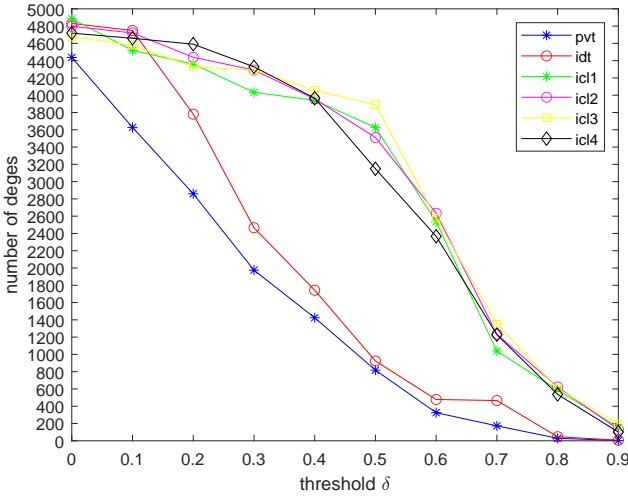

Figure 9: number of edges ranging from the $\delta$

be clearly seen that faithfulness can only be fully guaranteed when $\delta = 0$. However, such edges are not sparse enough to reflect some specific interpretable functions. When $\delta > 0.7$, it is almost impossible to recover to the original input, but the obtained skill graph can correspond well with previous methods. Additionally, in this paper, we default the $\delta$ for each skill to be PVT: 0.6, IDT: 0.7, ICL1-4: 0.8. Additionally, we have demonstrated in Figures 9 and 10 the changes in the number of edges and the continuous KL divergence metric with varying thresholds $\delta$. Specifically, Figure 9 presents the total number of edges in the circuit graph (excluding compensation circuit and bias circuit) under different thresholds, while Figure 10 shows the KL divergence between $\mathcal{G}^S$ and $\mathcal{G}*$ (solid lines) and $mathcalG^S$ and $\mathcal{G}$ (dash lines) obtained at different thresholds. Figure 9 clearly indicates that the edges with high causal effects from the previous token skill are the fewest, and the most are from the series of ICL skill, which corroborates the conclusion drawn from the clustering in Appendix E.2. Moreover, the changes in KL divergence (Figure 10) can be roughly divided into four phases (steady, burst, steady, burst). In conjunction with Figure 9, the two bursts are due to the rapid decrease in edges and the number of edges being too few, approaching zero. The default $\delta$ we selected (PVT 0.6, IDT 0.7, ICL1-4 0.8) are each in the second steady phase. Combining Figures 9 and10, it suggests that when a large number of edges are deleted, the circuit graph enters a phase of minimal change, which we believe best achieves the "balance between faithfulness and sparsity".

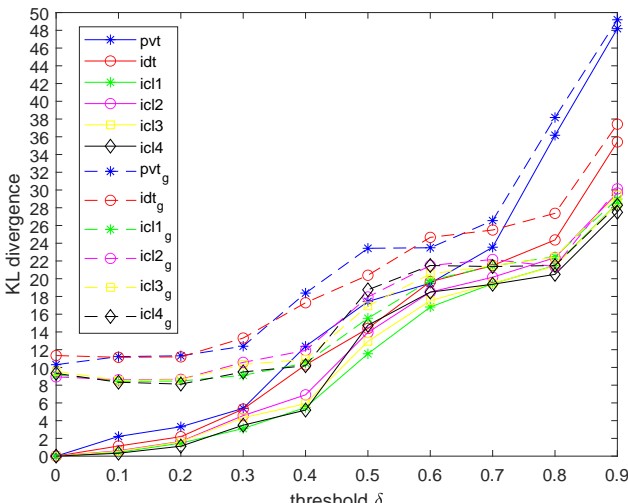

Figure 10: KL divergence ranging from the $\delta$, the solid lines represents KL between $\mathcal{G}^S$ and $\mathcal{G}*$, and the dash lines represents KL between $mathcalG^S$ and $\mathcal{G}$.

Table 10: Ratio of high $Eff$ path ($Eff > 0.5$) in $\mathcal{G}_{Bkg}*$ and $\mathcal{G}_{Self}*$ (The sum of ratios $> 1$ due to overlaps in each item).

| Skills | $\mathcal{G}_{Bkg}*$ | | | | | | | $\mathcal{G}_{Self}*$ | | | | | | |
| | $\mathcal{G}^S_{PVT}$ | $\mathcal{G}^S_{IDT}$ | $\mathcal{G}^S_{ICL1}$ | $\mathcal{G}^S_{ICL2}$ | $\mathcal{G}^S_{ICL3}$ | $\mathcal{G}^S_{ICL4}$ | Others | $\mathcal{G}^S_{PVT}$ | $\mathcal{G}^S_{IDT}$ | $\mathcal{G}^S_{ICL1}$ | $\mathcal{G}^S_{ICL2}$ | $\mathcal{G}^S_{ICL3}$ | $\mathcal{G}^S_{ICL4}$ | Others |
|---|---|---|---|---|---|---|---|---|---|---|---|---|---|---|
| Induction | 0.76 | - | - | - | - | - | 0.24 | 0.84 | - | - | - | - | - | 0.16 |
| ICL1 | 0.43 | 0.38 | 0.29 | 0.19 | 0.25 | 0.23 | 0.18 | 0.51 | 0.33 | 0.24 | 0.16 | 0.18 | 0.15 | 0.15 |
| ICL2 | 0.46 | 0.37 | 0.25 | 0.16 | 0.19 | 0.21 | 0.17 | 0.61 | 0.24 | 0.25 | 0.14 | 0.19 | 0.18 | 0.15 |
| ICL3 | 0.45 | 0.35 | 0.23 | 0.21 | 0.15 | 0.19 | 0.20 | 0.60 | 0.28 | 0.25 | 0.16 | 0.18 | 0.19 | 0.11 |
| ICL4 | 0.49 | 0.36 | 0.25 | 0.19 | 0.26 | 0.14 | 0.16 | 0.61 | 0.25 | 0.23 | 0.19 | 0.16 | 0.13 | 0.13 |

Additionally, we can observe that the KL divergence between $\mathcal{G}*$ and $\mathcal{G}$ is approximately 10 (as can be seen from the solid and dashed lines corresponding to $\delta = 0$), and generally, the KL divergence between $\mathcal{G}^S$ and $\mathcal{G}$ ($KL(\mathcal{G}^S, \mathcal{G})$) is greater than the KL divergence between $\mathcal{G}^S$ and $\mathcal{G}*$ ($KL(\mathcal{G}^S, \mathcal{G}*)$). Interestingly, as $\delta$ increases, the values between $KL(\mathcal{G}^S, \mathcal{G})$ and $KL(\mathcal{G}^S, \mathcal{G})$ get closer and are almost the same at the default threshold.

## F  DETAILS ABOUT VALIDATIONS FOR CAUSAL EFFECTS

Another question is whether the background effect and self effect, mentioned in Section 3.3, potentially exist as confounders or share the circuits with observed skills? To answer this question, we examine the paths in background/self text with $Eff > 0.5$. Table 11 categorizes these paths into 7 types and displays their ratios. Here, $\mathcal{G}^S_{PVT}$ signifies the ratio of those paths found in the Previous Token Skill graph, $\mathcal{G}^S_{IDT}$ refers to the ratio of those located in the Induction skill graph, similarly, $\mathcal{G}^S_{ICL1}$ to $\mathcal{G}^S_{ICL4}$ represents the ratio of paths in corresponding ICL skill graphs, and "Others" represents the ratio of paths that do not exist in either skill graphs. Notably, a small fraction of high-effect paths does not belong to any observed skill (approximately 0.1-0.2 in "Others"); these are the confounding paths we mentioned before. Additionally, we demonstrated the bivariate probability density function (PDF) in Figure 11. Bivariate PDF constructed from the origin text as one variable, and background text or self text as another one variable. Evidently, across all skills, the paths that have a high effect ($Eff > 0.5$) in the origin text include a part of paths with a relatively high effect ($Eff > 0.5$) in the background text. However, there are nearly ignorable high-effect paths in the self text in ICL skills. We guess that within the ICL skill, the background text and the origin text possess a significantly higher number of tokens compared to the self text, thereby leading to an insignificant effect of the self text.

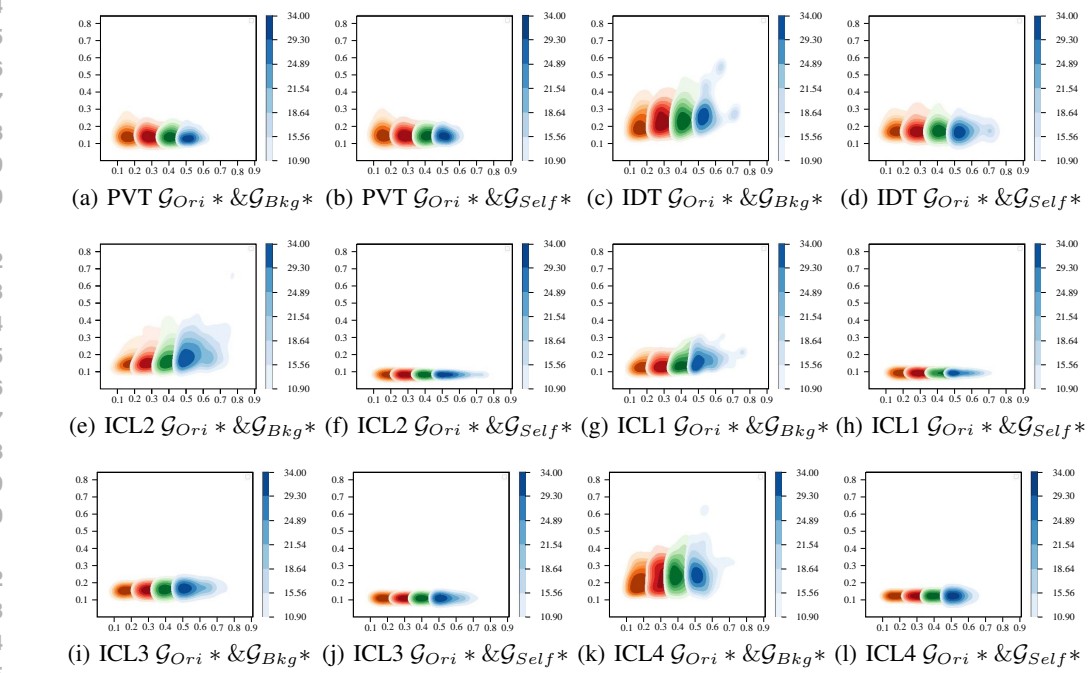

Figure 11: Bivariate probability density function (PDF) of path effects on Previous Token,Induction, ICL1 ICL2, ICL3, and ICL4 Skills. The x-axis represents the first variable, the path effect in the origin text ($\mathcal{G}_{Ori}*$) while the y-axis represents the second variable, the path effect in the background/self text ($\mathcal{G}_{Bkg}*/\mathcal{G}_{Self}*$). Orange, red, green, and blue respectively represent the distribution of paths with $Eff > 0.2, 0.3, 0.4, 0.5$ in the origin text.

Additionally, Table 11 also shows that a part of high-effect paths in the background/self text is common with the corresponding skill graph. Fortunately, we need not worry that removing these paths would render the final Skill Graph (paths) incomplete. Appendix G provides evidence that these removed but common paths can always be restored through multi-step paths (We explain this phenomenon as 'Inclusiveness' in Section 6.).

We have supplemented the bivariate distribution figures for Previous Token, ICL2, ICL3, and ICL4, as depicted in Figure 11.

# G   INCLUSIVE PATH

we have listed the whole paths for Previous Token Skills, all multi-step paths for the Induction and ICL1 Skills in following, with index of the send circuit, the first receive circuit, the second receive circuit.... The green represents the paths involving inclusive paths.

**Previous Token Skill**

*layer 0 circuit 13, layer 1 circuit 6, with effect 0.71*
*layer 0 circuit 14, layer 1 circuit 7, with effect 0.82*
*layer 0 circuit 16, layer 1 circuit 7, with effect 0.7*
*layer 0 circuit 20, layer 1 circuit 7, with effect 0.86*
*layer 0 circuit 14, layer 1 circuit 8, with effect 0.79*
*layer 0 circuit 16, layer 1 circuit 8, with effect 0.78*
*layer 0 circuit 17, layer 1 circuit 8, with effect 0.81*
*layer 0 circuit 19, layer 1 circuit 8, with effect 0.72*
*layer 0 circuit 20, layer 1 circuit 8, with effect 0.88*
*layer 0 circuit 22, layer 1 circuit 8, with effect 0.81*
*layer 0 circuit 23, layer 1 circuit 8, with effect 0.87*

*layer 0 circuit 24, layer 1 circuit 8, with effect 0.75*
*layer 0 circuit 13, layer 1 circuit 18, with effect 0.79*
*layer 0 circuit 13, layer 1 circuit 19, with effect 0.89*
*layer 0 circuit 14, layer 1 circuit 19, with effect 0.83*
*layer 0 circuit 15, layer 1 circuit 19, with effect 0.74*
*layer 0 circuit 16, layer 1 circuit 19, with effect 0.81*
*layer 0 circuit 20, layer 1 circuit 19, with effect 0.82*
*layer 0 circuit 24, layer 1 circuit 19, with effect 0.84*
*layer 0 circuit 13, layer 1 circuit 20, with effect 0.84*
*layer 0 circuit 14, layer 1 circuit 20, with effect 0.81*
*layer 0 circuit 20, layer 1 circuit 20, with effect 0.8*
*layer 0 circuit 13, layer 1 circuit 21, with effect 0.78*
*layer 0 circuit 14, layer 1 circuit 21, with effect 0.83*
*layer 0 circuit 16, layer 1 circuit 21, with effect 0.79*
*layer 0 circuit 17, layer 1 circuit 21, with effect 0.75*
*layer 0 circuit 20, layer 1 circuit 21, with effect 0.87*
*layer 0 circuit 22, layer 1 circuit 21, with effect 0.77*
*layer 0 circuit 23, layer 1 circuit 21, with effect 0.77*
*layer 0 circuit 24, layer 1 circuit 21, with effect 0.75*
*layer 0 circuit 23, layer 2 circuit 1, with effect 0.8*
*layer 0 circuit 24, layer 2 circuit 1, with effect 0.81*
*layer 1 circuit 13, layer 2 circuit 1, with effect 0.76*
*layer 1 circuit 15, layer 2 circuit 1, with effect 0.79*
*layer 1 circuit 16, layer 2 circuit 1, with effect 0.75*
*layer 1 circuit 17, layer 2 circuit 1, with effect 0.75*
*layer 1 circuit 20, layer 2 circuit 1, with effect 0.82*
*layer 0 circuit 13, layer 1 circuit 20, layer 2 circuit 1, with effect 0.74*
*layer 1 circuit 21, layer 2 circuit 1, with effect 0.8*
*layer 0 circuit 20, layer 1 circuit 21, layer 2 circuit 1, with effect 0.77*
*layer 1 circuit 22, layer 2 circuit 1, with effect 0.76*
*layer 1 circuit 23, layer 2 circuit 1, with effect 0.79*
*layer 1 circuit 24, layer 2 circuit 1, with effect 0.8*
*layer 0 circuit 20, layer 2 circuit 14, with effect 0.74*
*layer 0 circuit 21, layer 2 circuit 14, with effect 0.75*
*layer 0 circuit 22, layer 2 circuit 14, with effect 0.77*
*layer 0 circuit 23, layer 2 circuit 14, with effect 0.72*
*layer 0 circuit 24, layer 2 circuit 14, with effect 0.84*
*layer 1 circuit 13, layer 2 circuit 14, with effect 0.72*
*layer 1 circuit 15, layer 2 circuit 14, with effect 0.8*
*layer 1 circuit 16, layer 2 circuit 14, with effect 0.72*
*layer 1 circuit 17, layer 2 circuit 14, with effect 0.8*
*layer 1 circuit 18, layer 2 circuit 14, with effect 0.74*
*layer 1 circuit 20, layer 2 circuit 14, with effect 0.79*
*layer 1 circuit 21, layer 2 circuit 14, with effect 0.79*
*layer 0 circuit 14, layer 1 circuit 21, layer 2 circuit 14, with effect 0.71*
*layer 0 circuit 20, layer 1 circuit 21, layer 2 circuit 14, with effect 0.77*
*layer 1 circuit 22, layer 2 circuit 14, with effect 0.81*
*layer 1 circuit 23, layer 2 circuit 14, with effect 0.76*
*layer 1 circuit 24, layer 2 circuit 14, with effect 0.86*
*layer 0 circuit 13, layer 2 circuit 18, with effect 0.82*
*layer 1 circuit 13, layer 2 circuit 18, with effect 0.88*
*layer 0 circuit 19, layer 2 circuit 20, with effect 0.72*
*layer 0 circuit 20, layer 2 circuit 20, with effect 0.79*
*layer 0 circuit 21, layer 2 circuit 20, with effect 0.72*
*layer 0 circuit 22, layer 2 circuit 20, with effect 0.77*
*layer 1 circuit 19, layer 2 circuit 20, with effect 0.75*
*layer 1 circuit 20, layer 2 circuit 20, with effect 0.76*
*layer 1 circuit 21, layer 2 circuit 20, with effect 0.7*
*layer 1 circuit 22, layer 2 circuit 20, with effect 0.76*

*layer 1 circuit 23, layer 11 circuit 1, with effect 0.74*
*layer 1 circuit 24, layer 11 circuit 1, with effect 0.75*
*layer 2 circuit 24, layer 11 circuit 1, with effect 0.73*
*layer 4 circuit 23, layer 11 circuit 1, with effect 0.74*
*layer 0 circuit 24, layer 11 circuit 14, with effect 0.77*
*layer 1 circuit 13, layer 11 circuit 14, with effect 0.74*
*layer 1 circuit 16, layer 11 circuit 14, with effect 0.74*
*layer 1 circuit 24, layer 11 circuit 14, with effect 0.82*
*layer 2 circuit 13, layer 11 circuit 14, with effect 0.75*
*layer 2 circuit 16, layer 11 circuit 14, with effect 0.76*
*layer 2 circuit 24, layer 11 circuit 14, with effect 0.81*
*layer 3 circuit 13, layer 11 circuit 14, with effect 0.75*
*layer 3 circuit 16, layer 11 circuit 14, with effect 0.75*
*layer 3 circuit 24, layer 11 circuit 14, with effect 0.81*
*layer 4 circuit 13, layer 11 circuit 14, with effect 0.76*
*layer 4 circuit 24, layer 11 circuit 14, with effect 0.81*
*layer 5 circuit 24, layer 11 circuit 14, with effect 0.82*
*layer 6 circuit 16, layer 11 circuit 14, with effect 0.76*
*layer 6 circuit 24, layer 11 circuit 14, with effect 0.79*
*layer 7 circuit 24, layer 11 circuit 14, with effect 0.77*
*layer 8 circuit 24, layer 11 circuit 14, with effect 0.78*
*layer 9 circuit 24, layer 11 circuit 14, with effect 0.77*
*layer 10 circuit 24, layer 11 circuit 14, with effect 0.77*

**Multi-Step Paths in Induction Skill**

*layer 0 circuit 20, layer 2 circuit 14, layer 5 circuit 11, with effect 0.6*
*layer 0 circuit 21, layer 2 circuit 14, layer 5 circuit 11, with effect 0.6*
*layer 1 circuit 16, layer 2 circuit 14, layer 5 circuit 11, with effect 0.6*
*layer 1 circuit 18, layer 2 circuit 14, layer 5 circuit 11, with effect 0.6*
*layer 1 circuit 20, layer 2 circuit 14, layer 5 circuit 11, with effect 0.6*
*layer 1 circuit 21, layer 2 circuit 14, layer 5 circuit 11, with effect 0.6*
*layer 1 circuit 22, layer 2 circuit 14, layer 5 circuit 11, with effect 0.61*
*layer 0 circuit 13, layer 2 circuit 20, layer 5 circuit 11, with effect 0.6*
*layer 0 circuit 20, layer 2 circuit 14, layer 11 circuit 1, with effect 0.61*
*layer 0 circuit 21, layer 2 circuit 14, layer 11 circuit 1, with effect 0.63*
*layer 1 circuit 18, layer 2 circuit 14, layer 11 circuit 1, with effect 0.61*
*layer 1 circuit 20, layer 2 circuit 14, layer 11 circuit 1, with effect 0.61*
*layer 1 circuit 21, layer 2 circuit 14, layer 11 circuit 1, with effect 0.61*
*layer 1 circuit 22, layer 2 circuit 14, layer 11 circuit 1, with effect 0.63*

**Multi-Step Paths in ICL1 Skill**

*layer 0 circuit 13, layer 1 circuit 19, layer 3 circuit 11, with effect 0.81*
*layer 0 circuit 14, layer 1 circuit 19, layer 3 circuit 11, with effect 0.85*
*layer 0 circuit 15, layer 1 circuit 19, layer 3 circuit 11, with effect 0.84*
*layer 0 circuit 16, layer 1 circuit 19, layer 3 circuit 11, with effect 0.85*
*layer 0 circuit 21, layer 1 circuit 19, layer 3 circuit 11, with effect 0.82*
*layer 0 circuit 22, layer 1 circuit 19, layer 3 circuit 11, with effect 0.85*
*layer 0 circuit 23, layer 1 circuit 19, layer 3 circuit 11, with effect 0.84*
*layer 0 circuit 24, layer 1 circuit 19, layer 3 circuit 11, with effect 0.85*
*layer 0 circuit 13, layer 2 circuit 14, layer 3 circuit 11, with effect 0.81*
*layer 0 circuit 20, layer 2 circuit 14, layer 3 circuit 11, with effect 0.81*
*layer 0 circuit 21, layer 2 circuit 14, layer 3 circuit 11, with effect 0.83*
*layer 0 circuit 22, layer 2 circuit 14, layer 3 circuit 11, with effect 0.83*
*layer 1 circuit 20, layer 2 circuit 14, layer 3 circuit 11, with effect 0.81*
*layer 1 circuit 21, layer 2 circuit 14, layer 3 circuit 11, with effect 0.82*
*layer 1 circuit 22, layer 2 circuit 14, layer 3 circuit 11, with effect 0.83*
*layer 1 circuit 23, layer 2 circuit 14, layer 3 circuit 11, with effect 0.8*
*layer 0 circuit 13, layer 2 circuit 20, layer 3 circuit 11, with effect 0.86*

*layer 0 circuit 14, layer 2 circuit 20, layer 3 circuit 11, with effect 0.85*
*layer 0 circuit 15, layer 2 circuit 20, layer 3 circuit 11, with effect 0.81*
*layer 0 circuit 16, layer 2 circuit 20, layer 3 circuit 11, with effect 0.85*
*layer 0 circuit 17, layer 2 circuit 20, layer 3 circuit 11, with effect 0.85*
*layer 0 circuit 18, layer 2 circuit 20, layer 3 circuit 11, with effect 0.81*
*layer 0 circuit 19, layer 2 circuit 20, layer 3 circuit 11, with effect 0.82*
*layer 0 circuit 20, layer 2 circuit 20, layer 3 circuit 11, with effect 0.85*
*layer 0 circuit 21, layer 2 circuit 20, layer 3 circuit 11, with effect 0.83*
*layer 0 circuit 22, layer 2 circuit 20, layer 3 circuit 11, with effect 0.86*
*layer 0 circuit 24, layer 2 circuit 20, layer 3 circuit 11, with effect 0.81*
*layer 1 circuit 13, layer 2 circuit 20, layer 3 circuit 11, with effect 0.86*
*layer 1 circuit 14, layer 2 circuit 20, layer 3 circuit 11, with effect 0.84*
*layer 1 circuit 15, layer 2 circuit 20, layer 3 circuit 11, with effect 0.82*
*layer 1 circuit 16, layer 2 circuit 20, layer 3 circuit 11, with effect 0.85*
*layer 1 circuit 17, layer 2 circuit 20, layer 3 circuit 11, with effect 0.85*
*layer 1 circuit 18, layer 2 circuit 20, layer 3 circuit 11, with effect 0.85*
*layer 1 circuit 19, layer 2 circuit 20, layer 3 circuit 11, with effect 0.85*
*layer 0 circuit 14, layer 1 circuit 19, layer 2 circuit 20, layer 3 circuit 11, with effect 0.83*
*layer 0 circuit 15, layer 1 circuit 19, layer 2 circuit 20, layer 3 circuit 11, with effect 0.83*
*layer 0 circuit 16, layer 1 circuit 19, layer 2 circuit 20, layer 3 circuit 11, with effect 0.83*
*layer 0 circuit 22, layer 1 circuit 19, layer 2 circuit 20, layer 3 circuit 11, with effect 0.83*
*layer 0 circuit 23, layer 1 circuit 19, layer 2 circuit 20, layer 3 circuit 11, with effect 0.82*
*layer 0 circuit 24, layer 1 circuit 19, layer 2 circuit 20, layer 3 circuit 11, with effect 0.84*
*layer 1 circuit 20, layer 2 circuit 20, layer 3 circuit 11, with effect 0.85*
*layer 1 circuit 21, layer 2 circuit 20, layer 3 circuit 11, with effect 0.84*
*layer 1 circuit 22, layer 2 circuit 20, layer 3 circuit 11, with effect 0.86*
*layer 1 circuit 23, layer 2 circuit 20, layer 3 circuit 11, with effect 0.82*
*layer 1 circuit 24, layer 2 circuit 20, layer 3 circuit 11, with effect 0.81*
*layer 0 circuit 21, layer 2 circuit 14, layer 3 circuit 14, with effect 0.8*
*layer 0 circuit 22, layer 2 circuit 14, layer 3 circuit 14, with effect 0.81*
*layer 1 circuit 21, layer 2 circuit 14, layer 3 circuit 14, with effect 0.81*
*layer 1 circuit 22, layer 2 circuit 14, layer 3 circuit 14, with effect 0.81*
layer 0 circuit 13, layer 1 circuit 16, layer 10 circuit 9, with effect 0.84
layer 0 circuit 14, layer 1 circuit 16, layer 10 circuit 9, with effect 0.81
layer 0 circuit 15, layer 1 circuit 16, layer 10 circuit 9, with effect 0.8
layer 0 circuit 22, layer 1 circuit 16, layer 10 circuit 9, with effect 0.81
layer 0 circuit 14, layer 1 circuit 20, layer 10 circuit 9, with effect 0.83
layer 0 circuit 24, layer 1 circuit 20, layer 10 circuit 9, with effect 0.81
*layer 0 circuit 13, layer 2 circuit 20, layer 10 circuit 9, with effect 0.92*
*layer 0 circuit 14, layer 2 circuit 20, layer 10 circuit 9, with effect 0.9*
*layer 0 circuit 15, layer 2 circuit 20, layer 10 circuit 9, with effect 0.85*
*layer 0 circuit 16, layer 2 circuit 20, layer 10 circuit 9, with effect 0.91*
*layer 0 circuit 17, layer 2 circuit 20, layer 10 circuit 9, with effect 0.89*
*layer 0 circuit 18, layer 2 circuit 20, layer 10 circuit 9, with effect 0.86*
*layer 0 circuit 19, layer 2 circuit 20, layer 10 circuit 9, with effect 0.86*
*layer 0 circuit 20, layer 2 circuit 20, layer 10 circuit 9, with effect 0.9*
*layer 0 circuit 21, layer 2 circuit 20, layer 10 circuit 9, with effect 0.87*
*layer 0 circuit 22, layer 2 circuit 20, layer 10 circuit 9, with effect 0.92*
*layer 0 circuit 23, layer 2 circuit 20, layer 10 circuit 9, with effect 0.85*
*layer 0 circuit 24, layer 2 circuit 20, layer 10 circuit 9, with effect 0.86*
*layer 1 circuit 13, layer 2 circuit 20, layer 10 circuit 9, with effect 0.92*
*layer 1 circuit 14, layer 2 circuit 20, layer 10 circuit 9, with effect 0.89*
*layer 1 circuit 15, layer 2 circuit 20, layer 10 circuit 9, with effect 0.85*
*layer 1 circuit 16, layer 2 circuit 20, layer 10 circuit 9, with effect 0.9*
*layer 0 circuit 13, layer 1 circuit 16, layer 2 circuit 20, layer 10 circuit 9, with effect 0.83*
*layer 1 circuit 17, layer 2 circuit 20, layer 10 circuit 9, with effect 0.9*
*layer 1 circuit 18, layer 2 circuit 20, layer 10 circuit 9, with effect 0.91*
*layer 0 circuit 14, layer 1 circuit 18, layer 2 circuit 20, layer 10 circuit 9, with effect 0.81*
*layer 0 circuit 23, layer 1 circuit 18, layer 2 circuit 20, layer 10 circuit 9, with effect 0.83*

*layer 1 circuit 19, layer 2 circuit 20, layer 10 circuit 9, with effect 0.9*
*layer 0 circuit 13, layer 1 circuit 19, layer 2 circuit 20, layer 10 circuit 9, with effect 0.83*
*layer 0 circuit 14, layer 1 circuit 19, layer 2 circuit 20, layer 10 circuit 9, with effect 0.87*
*layer 0 circuit 15, layer 1 circuit 19, layer 2 circuit 20, layer 10 circuit 9, with effect 0.86*
*layer 0 circuit 16, layer 1 circuit 19, layer 2 circuit 20, layer 10 circuit 9, with effect 0.87*
*layer 0 circuit 20, layer 1 circuit 19, layer 2 circuit 20, layer 10 circuit 9, with effect 0.82*
*layer 0 circuit 21, layer 1 circuit 19, layer 2 circuit 20, layer 10 circuit 9, with effect 0.82*
*layer 0 circuit 22, layer 1 circuit 19, layer 2 circuit 20, layer 10 circuit 9, with effect 0.87*
*layer 0 circuit 23, layer 1 circuit 19, layer 2 circuit 20, layer 10 circuit 9, with effect 0.86*
*layer 0 circuit 24, layer 1 circuit 19, layer 2 circuit 20, layer 10 circuit 9, with effect 0.88*
*layer 1 circuit 20, layer 2 circuit 20, layer 10 circuit 9, with effect 0.9*
*layer 0 circuit 14, layer 1 circuit 20, layer 2 circuit 20, layer 10 circuit 9, with effect 0.81*
*layer 1 circuit 21, layer 2 circuit 20, layer 10 circuit 9, with effect 0.89*
*layer 1 circuit 22, layer 2 circuit 20, layer 10 circuit 9, with effect 0.92*
*layer 1 circuit 23, layer 2 circuit 20, layer 10 circuit 9, with effect 0.86*
*layer 0 circuit 14, layer 1 circuit 19, layer 10 circuit 10, with effect 0.81*
*layer 0 circuit 16, layer 1 circuit 19, layer 10 circuit 10, with effect 0.81*
layer 0 circuit 22, layer 1 circuit 19, layer 10 circuit 10, with effect 0.81
layer 0 circuit 23, layer 1 circuit 19, layer 10 circuit 10, with effect 0.81
*layer 0 circuit 24, layer 1 circuit 19, layer 10 circuit 10, with effect 0.82*
*layer 0 circuit 14, layer 1 circuit 19, layer 11 circuit 5, with effect 0.81*
*layer 0 circuit 16, layer 1 circuit 19, layer 11 circuit 5, with effect 0.8*
layer 0 circuit 22, layer 1 circuit 19, layer 11 circuit 5, with effect 0.81
*layer 0 circuit 24, layer 1 circuit 19, layer 11 circuit 5, with effect 0.81*
layer 0 circuit 13, layer 2 circuit 14, layer 11 circuit 5, with effect 0.87
layer 0 circuit 14, layer 2 circuit 14, layer 11 circuit 5, with effect 0.81
*layer 0 circuit 20, layer 2 circuit 14, layer 11 circuit 5, with effect 0.86*
*layer 0 circuit 21, layer 2 circuit 14, layer 11 circuit 5, with effect 0.89*
*layer 0 circuit 22, layer 2 circuit 14, layer 11 circuit 5, with effect 0.89*
*layer 0 circuit 23, layer 2 circuit 14, layer 11 circuit 5, with effect 0.86*
*layer 0 circuit 24, layer 2 circuit 14, layer 11 circuit 5, with effect 0.84*
*layer 1 circuit 13, layer 2 circuit 14, layer 11 circuit 5, with effect 0.85*
layer 1 circuit 14, layer 2 circuit 14, layer 11 circuit 5, with effect 0.86
*layer 1 circuit 15, layer 2 circuit 14, layer 11 circuit 5, with effect 0.85*
*layer 1 circuit 16, layer 2 circuit 14, layer 11 circuit 5, with effect 0.84*
*layer 1 circuit 17, layer 2 circuit 14, layer 11 circuit 5, with effect 0.85*
*layer 1 circuit 18, layer 2 circuit 14, layer 11 circuit 5, with effect 0.86*
layer 1 circuit 19, layer 2 circuit 14, layer 11 circuit 5, with effect 0.8
*layer 1 circuit 20, layer 2 circuit 14, layer 11 circuit 5, with effect 0.87*
layer 1 circuit 21, layer 2 circuit 14, layer 11 circuit 5, with effect 0.89
*layer 1 circuit 22, layer 2 circuit 14, layer 11 circuit 5, with effect 0.89*
*layer 1 circuit 23, layer 2 circuit 14, layer 11 circuit 5, with effect 0.86*
*layer 1 circuit 24, layer 2 circuit 14, layer 11 circuit 5, with effect 0.81*
layer 0 circuit 13, layer 2 circuit 24, layer 11 circuit 5, with effect 0.84
layer 0 circuit 14, layer 2 circuit 24, layer 11 circuit 5, with effect 0.82
layer 0 circuit 15, layer 2 circuit 24, layer 11 circuit 5, with effect 0.85
layer 0 circuit 16, layer 2 circuit 24, layer 11 circuit 5, with effect 0.85
layer 0 circuit 17, layer 2 circuit 24, layer 11 circuit 5, with effect 0.85
layer 0 circuit 22, layer 2 circuit 24, layer 11 circuit 5, with effect 0.85
layer 0 circuit 23, layer 2 circuit 24, layer 11 circuit 5, with effect 0.85
layer 0 circuit 24, layer 2 circuit 24, layer 11 circuit 5, with effect 0.82
layer 1 circuit 13, layer 2 circuit 24, layer 11 circuit 5, with effect 0.83
layer 1 circuit 14, layer 2 circuit 24, layer 11 circuit 5, with effect 0.81
layer 1 circuit 15, layer 2 circuit 24, layer 11 circuit 5, with effect 0.82
layer 1 circuit 16, layer 2 circuit 24, layer 11 circuit 5, with effect 0.81
layer 1 circuit 17, layer 2 circuit 24, layer 11 circuit 5, with effect 0.81
layer 1 circuit 22, layer 2 circuit 24, layer 11 circuit 5, with effect 0.85
layer 1 circuit 23, layer 2 circuit 24, layer 11 circuit 5, with effect 0.82
layer 1 circuit 24, layer 2 circuit 24, layer 11 circuit 5, with effect 0.81

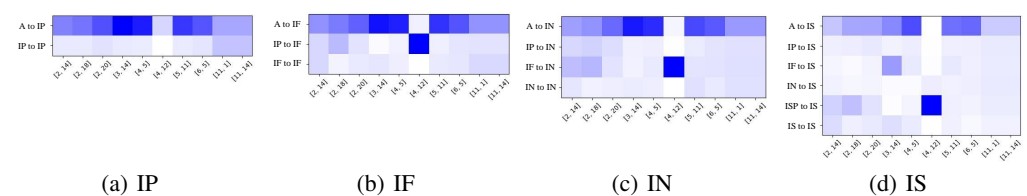

(a) IP          (b) IF          (c) IN          (d) IS

Figure 12: Attention weights of located token positions in Induction Skill

*layer 0 circuit 13, layer 3 circuit 14, layer 11 circuit 5, with effect 0.81*
*layer 0 circuit 23, layer 3 circuit 14, layer 11 circuit 5, with effect 0.85*
*layer 1 circuit 23, layer 3 circuit 14, layer 11 circuit 5, with effect 0.81*
*layer 2 circuit 23, layer 3 circuit 14, layer 11 circuit 5, with effect 0.8*

Almost all 3-step paths are composed of paths from lower-level skills. For instance, in the ICL skill, the sequence *"layer 0 circuit 20, layer 2 circuit 14, layer 5 circuit 11"* encompasses the path *"layer 0 circuit 20, layer 2 circuit 14"* from the previous token skill. Furthermore, it is apparent that the more complex a skill, the more multi-step paths it encompasses.

## H    ATTENTION WEIGHTS OF KEY CIRCUIT

In this section, we provide additional information on the attention weights of key circuits in the Induction Skill and ICL1 Skill.

For the Induction samples, we focus on the following tokens:

*"A ... IP IF IN ... ISP IS"*, where *"A"* represents the first token of the input text, *"IF"* and *"IS"* denote the positions of the first and second appearances of the duplicated token respectively, *"IP"* and *"IN"* indicate the tokens before and after *"IF"*, and *"ISP"* refers to the token before *"IS"*. Figure 12 shows these located positions' attention weight.

For the ICL samples, we select ICL1 (icl_sst2 task) to show, following tokens:

*"A B ... P1P P1A ... P1B L1... A2 ... P2P P2A ... P2B L2 ... A3 ... P3P P3A ... P3B*, where "A", "A2", "A3" represents the beginning of review1, review2 and review3, "P1P", "P2P", and "P3P" represents the end of review1, review2, and review3, "P1A ... P1B", "P2A ... P2B", "P3A ... P3B" represents the label prompt of review1, review2, and review3, and "L1", "L2" represent the label of review1 and review2. Figure 13 shows these located positions' attention weight.

## I    COMPARISONS WITH OTHER METHODS VALIDATING CONJECTURES

One of the contributions of this paper is to validate three long-unverified conjectures about language skills: Identifiability, Stratification, and Inclusiveness. The question of why previous work merely "proposed conjectures" while our method could find "strong evidence" will be answered in this section.

Firstly, we compared the differences in circuit components among 4 circuit discovery methods, including ours. The other methods are ACDC, OPT-prun, and EAP (introduced in Section 2). Each method used its own circuit discovery strategy to search for corresponding circuit graphs for the three skills we focus on: PVT, IDT, and ICL1. Then, as with Table 4, we investigated the distribution of receiver nodes in these circuit graphs and displayed the normalized results in Figure 14.

It is clear from other methods that PVT is more prominent in the shallow layers, IDT in the mid-to-deep layers, and ICL1 tends to cluster in the deep layers. However, the circuit graphs from these methods are insufficient to prove these patterns. For instance, although PVT is significantly concentrated in the shallow layers, there are also components in the deep layers. Yet, the circuit graph discovered by our method provides a more distinct differentiation: PVT circuits only appear in layers 1 and 2, and IDT circuits only appear in layers 1-6. Our method determines the specific layer numbers of the skill circuits and confirms that as the skill becomes more complex, the layers spread

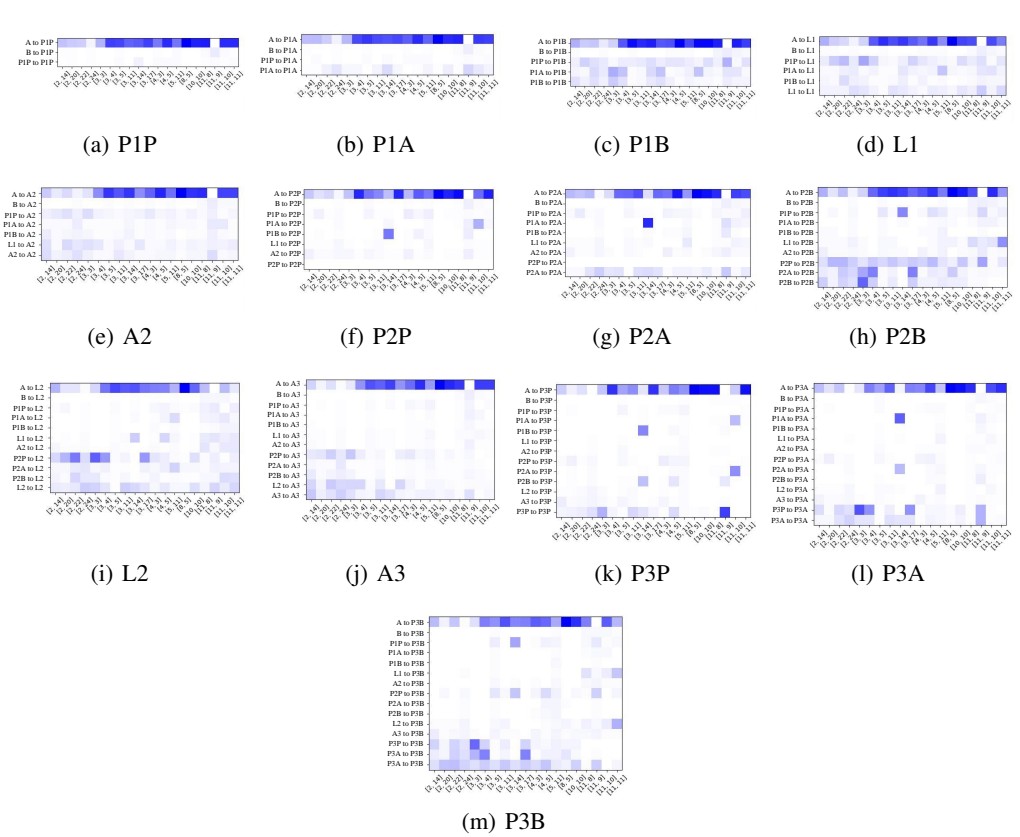

Figure 13: Attention weights of located token positions in ICL Skill

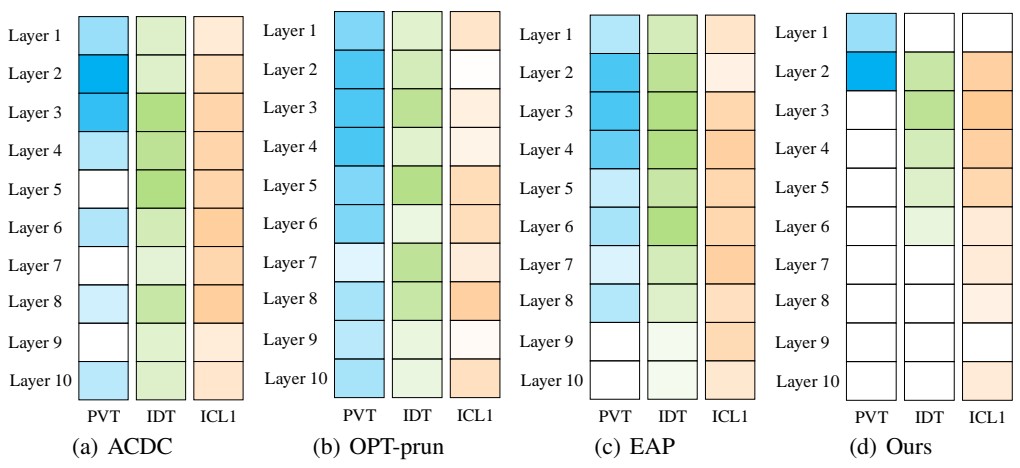

Figure 14: Visualization of receivers distributed in layer1-10 in 3 increasingly-complex skills (PVT, IDT, and ICL1), obtained from 4 circuit discovery methods (ACDC, OPT-prun, EAP, and Ours)

Table 11: Overlaps between different skill circuit graphs

| Method | PVT | | IDT | | ICL1 | |
|---|---|---|---|---|---|---|
| | $ovlp(PVT, IDT)$ | $ovlp(PVT, ICL1)$ | $ovlp(IDT, PVT)$ | $ovlp(IDT, ICL1)$ | $ovlp(ICL1, PVT)$ | $ovlp(ICL1, IDT)$ |
| ACDC | 0.13 | 0.05 | 0.19 | 0.10 | 0.06 | 0.17 |
| OPT-prun | 0.11 | 0.18 | 0.05 | 0.07 | 0.14 | 0.17 |
| EAP | 0.09 | 0.06 | 0.14 | 0.05 | 0.03 | 0.18 |
| Ours | 0.34 | 0.29 | 0.74 | 0.35 | 0.81 | 0.63 |

from shallow to deep. This finding provides stronger evidence for the identifiability and stratification of skills compared to other methods.

Additionally, to observe the performance of these methods on the conjecture of Inclusiveness, we investigated their overlap on the three skill circuits: PVT, IDT, and ICL1. The corresponding circuit graphs are still derived from the circuit discovery strategies proposed by each method, searching in the corpora corresponding to the three skills proposed in this paper. The rule for calculating overlap is as follows: let $ovlp(A, B)$ represent what the rate of edges in skill graph $A$ also existing in skill graph $B$ is. For any edge $e^i$ in skill graph $A$, we set an overlap flag $f_{A,B}(e^i)$. If $e^i$ in $A$ also exists in skill circuit graphs $B$, then $f_{A,B}(e^i) = 1$, otherwise $f_{A,B}(e^i) = 0$. For a circuit graph $A$ with $N_A$ edges, its set of edges is $\mathcal{E}_A$. Our overlap is calculated as $ovlp(A, B) = \frac{1}{N_A} \sum_{e^i \in \mathcal{E}_A}^{\mathcal{E}_A} f_{A,B}(e^i)$.

Table 5 demonstrates that the overlap of circuit graphs discovered by existing methods is quite low. For instance, $ovlp(ICL1, IDT)$ is only 0.17 in ACDC. However, this 0.17 overlap of circuits represents the key function of induction (often referred to as the **induction head**). As a result, many studies have proposed the conjecture that the ICL skill includes the Induction skill. Yet, only our work provides clear empirical evidence for the conjecture of inclusiveness: $ovlp(IDT, PVT) = 0.74$ indicates that 74% of the paths in the circuit graph of the Induction skill exist in the circuit graph of the previous token skill. Furthermore, $ovlp(ICL1, PVT) = 0.81$ and $ovlp(ICL1, IDT) = 0.63$ suggest that 81% and 63% of the paths in the ICL skill's circuit graph are included in the circuit graphs of the previous token skill and the induction skill, respectively.

## J    SKILL CIRCUIT GRAPHS

Due to large size constraints, we have only displayed the circuit graph for the Previous Token Skill. For additional skill graphs, please refer to our repository.

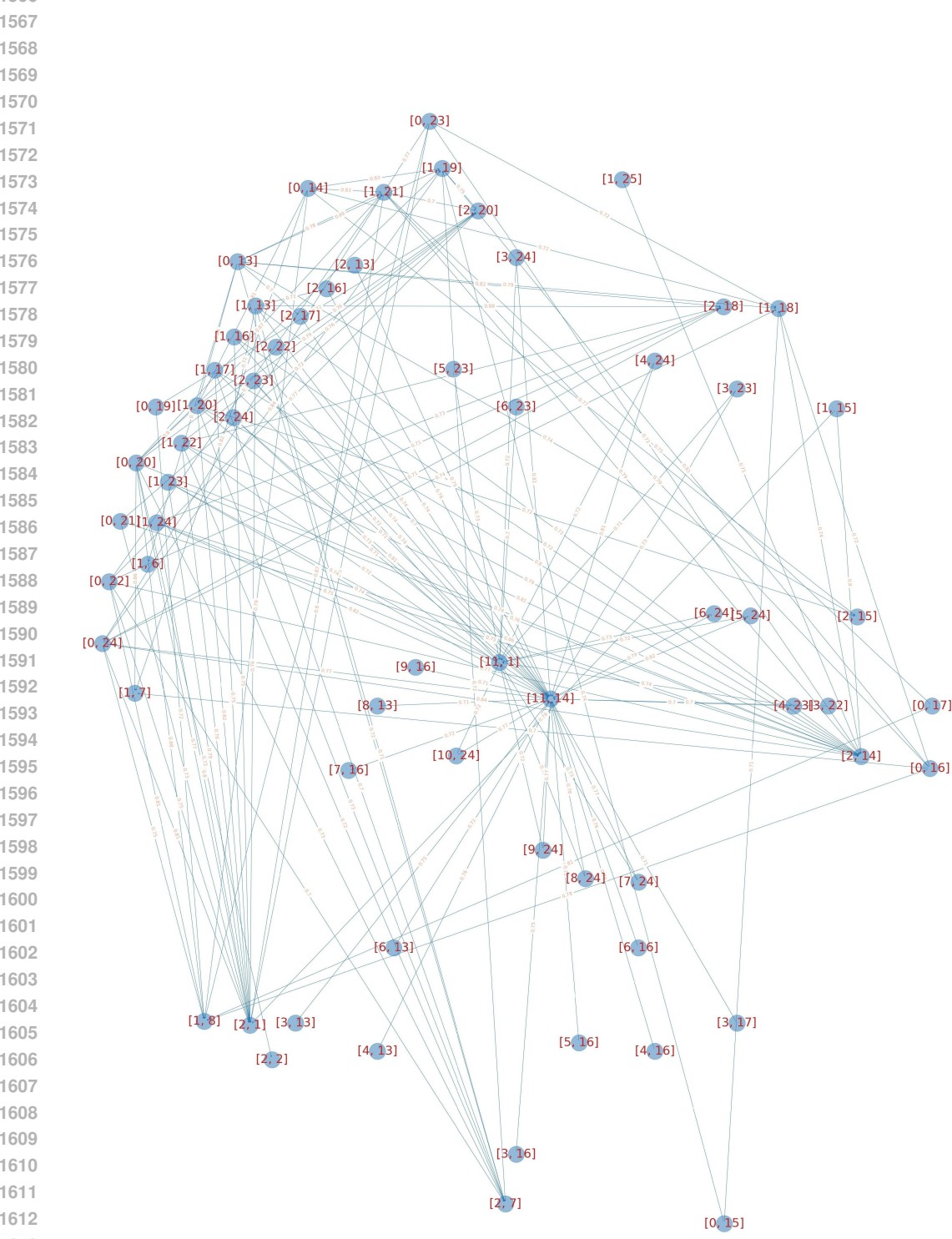

Figure 15: Skill Circuit Graph of Previous Token Skill, all paths with $Eff > 0.7$ are labeled.

