# OpenReview forum: "Unveiling Language Skills under Circuits"
_ICLR.cc/2025/Conference — ICLR 2025 Conference Withdrawn Submission_

### Official Review · Reviewer_HGAK · 2024-10-21

**Soundness:** 2
**Presentation:** 3
**Contribution:** 2
**Rating:** 3
**Confidence:** 3

**Summary:**

This paper introduces a novel framework for analyzing language model circuits, focusing on three language skills: previous token prediction, induction, and in-context learning.

The authors propose "Memory Circuits" as a new unit of analysis by rewriting of the Transformer layer and isolating the different computational path in a given layer. It enable the differentiation of the effects of the MLP alone, attention layer alone and MLP-attention layer interactions.

Using a three-step approach combining circuit decomposition, greedy path pruning, and comparative analysis of the paths contained in circuits from inputs without the studied pattern and inputs with the studied pattern, they identify circuit paths associated with different language skills.

Their analysis suggests that language skills are identifiable through circuit dissection, stratified across model layers, and build upon each other hierarchically.

**Strengths:**

- **Novel decomposition approach.** The introduction of Memory Units as a per-layer rewriting offers a new way to decompose neural networks, complementing existing feature-based analysis approaches. This could be particularly valuable for isolating the role of attention-MLP interactions within individual layers. The introduction of new unit of analysis is an exciting research direction, given that finding the right unit could unlock multiple analisys downstream.
- **Thoughtful experimental design for counterfactual inputs.** The use of background and self-text as counterfactual inputs demonstrates a sophisticated approach to isolating skill-specific effects from confounding factors like the effect of bigram statistics.
- **Hierarchical analysis of language skills.** The study provides evidence for the hierarchical nature of language model capabilities, showing how complex skills like in-context learning build upon simpler ones like previous token prediction.

**Weaknesses:**

- **Limited comparison with existing work.** The paper lacks discussion of how their approach relates to or improves upon existing circuit discovery techniques [1,2,3]. Given the number of existing techniques and the absence of ground-truth in the field of mechanistic interpretabilty, the authors should provide motivated arguments for where their technique adds value, or validate their technique of circuit identification on well-studied circuit. This is particularly important for their path decomposition approach, as similar ideas (including MLP analysis) have been explored in previous work [4,5].
- **Methodological limitations in circuit validation and identification:**
    - The greedy search algorithm relies solely on maintaining the top token prediction when pruning edges. This is a limited metric as it doesn't account for fine-grained differences in the output probability distribution and is highly non-linear due to the final softmax. As discussed in section 4 of the reference [7], the choice of metrics for causal intervention significantly impacts the identification of components contributing to model behavior.
    - The validation through ablation studies is insufficient. While ablating components can disable a given output, this approach may obscure the role of earlier components that the final components relied on to compute their values. More extensive validation techniques have been developed and discussed in the literature [6]. On top of computed validation metrics, one often rely on detailed investigation of the components, or on downstream applications (see [2] for instance) as additional ways to validate their findings. None of these ways is substantially present in this work.
    - The decomposition doesn't account for token position, which is a significant limitation given that most existing circuit analysis work have demonstrated widely different roles for components depending on their position of activation.
- **Technical imprecisions:**
    - The paper incorrectly identifies their model as GPT2-XL when describing an architecture with 12 layers and 12 attention heads, which actually corresponds to GPT2-small. See Table 2 of the original GPT-2 paper for reference [8].
    - There's improper use of tensor notation for the MLP layer in equation (1), attempting to encode non-linearity in a matrix through tensor products, which can only represent linear operations.
    - The paper claims that the greedy search finds the "smallest, independent, and functionally complete circuit graph," when greedy search cannot guarantee minimal solutions.
    - There's inconsistency between claims of a "lossless circuit framework" in the introduction and later acknowledgment of approximation errors. The paper only reports the minimum squared error between circuit sum and original output, leaving questions about average/median errors and the source of these approximations (mathematical reformulation vs. discrepancies in floating-point implementation).

[1] Goldowsky-Dill et al., "Localizing Model Behavior with Path Patching"

[2] Marks et al., "Sparse Feature Circuits: Discovering and Editing Interpretable Causal Graphs in Language Models"

[3] Conmy et al., "Towards Automated Circuit Discovery for Mechanistic Interpretability"

[4] Hanna et al., "How does GPT-2 compute greater-than?: Interpreting mathematical abilities in a pre-trained language model"

[5] Lieberum et al., "Does Circuit Analysis Interpretability Scale? Evidence from Multiple Choice Capabilities in Chinchilla"

[6] Miller et al., "Transformer Circuit Faithfulness Metrics Are Not Robust"

[7] Heimersheim et al., "How to use and interpret activation patching"

[8] Radford et al., "Language Models are Unsupervised Multitask Learners"

**Questions:**

1. How does your Memory Circuit decomposition provide specific advantages over existing path decomposition techniques, particularly for analyzing MLP-attention interactions?
2. How would your method and other circuit discovery techniques compare (e.g., [1,2,3]) on a common task?
3. Have you considered extending your analysis to account for token position?
4. Could you clarify the model (GPT-2 XL vs GPT-2 small) used in your experiments and update the name of the model or its architecture if a different model was used?
5. What is the distribution (mean, median, max) of approximation errors in your circuit decomposition, and what are the primary sources of these errors?
6. Could you provide a more mathematically precise formulation of the MLP decomposition that properly handles non-linearities?
7. For the in-context learning tasks analyzed, what is the base performance of GPT-2 on ground truth data?

---

> ### Author Response · Authors · 2024-11-21
> **response**
>
> Thank you for your thorough review and valuable suggestions, especially regarding the comparison with related work. As a result, we recognized these shortcomings and made revisions in the revised version. We apologize for our previous unclear presentation, which made our motivation and contributions unclear. Some major changes that can address your concerns are as follows:
>
> 1. In the introduction and related work sections, we emphasized the **research gap** in existing work, namely the **unfaithfulness** to the output, and provided ample empirical evidence and analysis.
> 2. To address the gap in point 1, we proposed a **new pruning strategy**. To our knowledge, this is the only pruning strategy to date that can ensure output faithfulness (you can see an overview in the revised introduction section, with detailed content in Sections 3.2 and 3.3).
> 3. In addition, our method has other advantages, such as solving the **non-linearity of MLP** in circuit decoupling (we answer this question in detail in the subsequent responses).
>
> Below are our responses to your questions:

---

> > ### Author Response · Authors · 2024-11-21
> >
> > ### Q1How does your Memory Circuit decomposition provide specific advantages over existing path decomposition techniques, particularly for analyzing MLP-attention interactions?
> >
> > Overall, A more precise statement would be: **we decompose the language model into memory circuits, compensation circuits, and bias circuits**. Among them, memory circuits can be linearly combined, while compensation circuits represent the loss when the non-linear MLP is decomposed into a linear combination.
> >
> > Specifically, for a nonlinear function $f(·)$, we have $f(A+B)\neq f(A)+f(B)$. By analogy, the input to the MLP can be seen as coming from two sources: the residual stream from the previous layer and the output of the attention (together they form the residual stream). However, due to the nonlinearity of the MLP, **the impact of these two on the MLP cannot be represented independently**. This is the functional irreversibility between the language model architecture and the circuit graph. Therefore, we use a **compensation circuit**, i.e., $f(A+B)= f(A)+f(B)+Cps(A,B)$, so $Cps(A,B)=f(A+B)-f(A)-f(B)$. In this way, **deleting $f(A)$ alone will not affect $f(B)$**, thus ensuring the lossless decoupling of the circuit.
> >
> > Additionally, regarding the understanding of memory circuits, we define them as the smallest units for reading memory. Under the condition that the compensation circuit ensures linear decoupling, each memory circuit can reflect an operation of LLM reading memory, which we consider as a basic operation of reasoning (you can refer to our detailed explanation about reading memory and reasoning in **Appendix B**).

---

> > ### Author Response · Authors · 2024-11-21
> >
> > ### Q2 How would your method and other circuit discovery techniques compare (e.g., [1,2,3]) on a common task?
> >
> > We have added a comparison with these methods in the related work (**Section 2**). Overall, we compared these methods across three common tasks (IOI, greater-than, and induction). An important conclusion we found is that the pruning strategies proposed by these methods cannot guarantee that the final circuit output is consistent with the original LLM output (the **unfaithfulness of output** we mentioned before). Theoretically, this is possible, as the pruning strategies of existing methods are dedicated to minimizing logit changes for counterfactuals, but after a large number of edges are pruned, the **accumulated bias** will cause changes in the final output.
> >
> > Our pruning method, on the other hand, ensures the faithfulness of the circuit output. Therefore, compared with previous circuit pruning methods, we have two advantages: 1, existing pruning strategies change the results of reasoning, while our pruning strategy ensures **faithfulness** to the original LLM output. 2, Moreover, given that we proposed a fully-linear decomposition, the existing circuit decomposition is non-linear thus functionally irreversible, while our **lossless and linear** decomposition ensures the functional reversibility of the LLM structure and circuit graph conversion.
> >
> >
> >
> >
> > [1] Goldowsky-Dill et al., "Localizing Model Behavior with Path Patching"
> >
> > [2] Marks et al., "Sparse Feature Circuits: Discovering and Editing Interpretable Causal Graphs in Language Models"
> >
> > [3] Conmy et al., "Towards Automated Circuit Discovery for Mechanistic Interpretability"

---

> > ### Author Response · Authors · 2024-11-21
> >
> > ### Q4 Could you clarify the model (GPT-2 XL vs GPT-2 small) used in your experiments and update the name of the model or its architecture if a different model was used?
> >
> > This was an oversight in expression, which we have now corrected.

---

> > ### Author Response · Authors · 2024-11-21
> >
> > ### Q5 What is the distribution (mean, median, max) of approximation errors in your circuit decomposition, and what are the primary sources of these errors?
> >
> > The source of these errors is the **floating-point computation in Torch**, meaning that only the least bit will differ, which is actually **negligible**. It is only from an absolutely rigorous perspective that we refer to it as "nearly accurate".

---

> > ### Author Response · Authors · 2024-11-21
> >
> > ### Q6 Could you provide a more mathematically precise formulation of the MLP decomposition that properly handles non-linearities?
> >
> > We understand your concerns. We have now added an explanation for the compensation circuit in the **Appendix C** and explained that Equation 1 is derived under the condition of **ignoring** the compensation circuit, hence it is linear.

---

> > ### Author Response · Authors · 2024-11-21
> >
> > ### Q7 For the in-context learning tasks analyzed, what is the base performance of GPT-2 on ground truth data?
> >
> > All our data are sampled under the condition that the LLM **answers correctly** (see **Appendix E**). This is reasonable because for samples that the LLM cannot answer correctly, we cannot confirm that it has executed the corresponding skill.

---

> ### Author Response · Authors · 2024-11-21
>
> ### Q3 Have you considered extending your analysis to account for token position?
>
> We are sorry but this is **out of the scope of our work**. We are aware that some existing works have focused on QK circuits, which provide fine-grained insights into token positions. However, the language skill we focus on is a coarse-grained level behavior, which is a **common** rule obtained under **sufficient samples**. Currently, we have not yet developed a reverse engineering method to obtain language skills from a single sample.

---

> ### Comment · Reviewer_HGAK · 2024-11-23
> **General response**
>
> Thank you for your comprehensive answer, I appreciate the significant modification you performed on the paper. However, they don't address my core methodological concern about the work (see the comment following Q2).
>
> **Q2 - Comparison with other works**
>
> Thank you for including a comparison with other work, this is a step in the right direction to be able to assess the significance of your contribution.
>
> However, the comparison presented reflect a misunderstanding of the metrics used for circuit analysis.
>
> *Discrete Metric choice*. You are using only the token output with maximal prediction as the metric. However this metric is the result of applying a discrete mapping on the continuous logits outputted by the model. The 'output faithfulness' ensured by the greedy search perform is only an artefact of this discrete mapping. The output logits are changing after each edge removal in Algorithm 1, but not enough to cross the threshold to change the identity of the max logit. To measure circuit faithfulness, typical work of circuit analysis focus on *continuous* metrics like KL divergence of the post-softmax output of the original model and the circuit. Logit difference is also commonly used. See the section 4 of "How to use and interpret activation patching" by Heimersheim and Nanda for a detailed description of the merit of different metrics for circuit analysis, and the importance of continuous metrics.
>
> *Circuit analysis is a tradeoff between the faithfulness of the circuit and its sparseness*. The full model is a perfectly faithful but useless circuit, as it doesn't give any insight on the component used for solving the task. On the other end, a circuit made of very few edges and component is very sparse, easy to map and study, but is likely to be unfaithful as model behaviors are typically distributed across different part of the model (even if some component are more involved than other).
>
> To capture this tradeoff, circuit analysis works typically include curves with varying threshold to show that a given technique provide Pareto improvement compared to other techniques, i.e. for a given circuit size, the technique is able to find a circuit that capture more of the behavior than alternatives. See for instance Figure 4 of "Towards Automated Circuit Discovery for Mechanistic Interpretability" for an example of such curve.
>
> Your comparison from section 1 doesn't take into consideration this fondamental tradeoff. in particular it doesn't compare the size of the circuits considered.
>
> **Q6 - Presentation of equation 1**
>
> As it stands, the subsection 3.1 starts with a misleading and equation (1) "we propose a complete decomposition of the transformer model including the MLP layers. Using tensor products (⊗), we can represent any layer of the transformer model:". However, as noted in the original review, you use a linear notation to represent the non-linear MLP layer.
>
> I understand the role of the compensation circuit, however, they are only mentioned at line 181 in this subsection. They should be included in equation (1) from the start.
>
>
> **Q7 - Choice of data**
>
> Original question: For the in-context learning tasks analyzed, what is the base performance of GPT-2 on ground truth data?
>
> > All our data are sampled under the condition that the LLM **answers correctly** (see **Appendix E**). This is reasonable because for samples that the LLM cannot answer correctly, we cannot confirm that it has executed the corresponding skill.
>
> For samples where the LLM answer correctly, we cannot be sure that the LLM has executed the corresponding skill neither.
>
> Imagine a model that is supposed to perform addition of two number x and y in [0,50].
>
> In fact, the model internally creates a pseudo random number between 0 and 100 for each pair (x,y). If I select only the pair (x,y) where the model is correct, this doesn't mean the model is performing addition for these samples.
>
> Instead, to be sure my model has the proper skill (or at least, a reasonably degraded version of the skill), I need to check its accuracy on a representative dataset. Then, it doesn't matter if I include samples where the model gets it correct or not. I assume that in general the model possess a good enough version on the skill in general.

---

> > ### Author Response · Authors · 2024-11-24
> > **thank you for your response！**
> >
> > Thank you for your response. Your response is both insightful and professional, precisely the kind of discuss we anticipate! However, there are still significant misunderstandings about our work, which we will clarify below.
> >
> > ### Q2: Faithfulness and Sparsity of Circuits
> >
> > We completely agree with your views on faithfulness and sparsity, (especially the tradeoff between them,) which align with our thoughts, but you have **misunderstood** our work.
> >
> > Perhaps you can consider the issue this way: the bias of the circuit graph's unfaithfulness comes from **two parts**, the first part is the **counterfactual operation** (because there is a threshold), and the second part is the **pruning metric** (as we said, many metrics will have cumulative bias). Therefore, what you referred to as "the tradeoff between the faithfulness of the circuit and its sparseness" actually refers to the **balance between the threshold of the counterfactual and the final rank of the token**. What we want to solve is the bias brought by the second part, the pruning metric. It is known that conducting counterfactuals and maximal prediction **simultaneously** will produce **very poor results** (as clarified in Section 4 of "How to use and interpret activation patching" that you mentioned), so we adopt a method that **completely separates** pruning and causal analysis.
> >
> > Compared to existing pruning work, our pruning work is divided into **two steps**: the first step is fidelity pruning, resulting in the irreducible circuit graph$G*$, and the second step is causal analysis, which applies causal corrections to $G*$, resulting in the skill circuit graph$G^{S}$ (existing work directly obtains $G^{S}$). However, you persistently misinterpret our $G*$ as being comparable to others' $G^{S}$. The issues you mentioned, such as "*You are using only the token output with maximal prediction as the metric...,*" are all settings for our $G*$, but you criticize it as if it were $G^{S}$. Moreover, the section 4 you mentioned in "How to use and interpret activation patching" directly uses the **Rank of token** as an optimization to obtain experimental results for $G^{S}$, which does not prove that it is **unreasonable** for us to apply Rank of token to $G*$.
> >
> > In addition, we have conducted counterfactual and intervention analyses on $G*$. Under these causal analyses, $G^{S}$ also includes those circuit paths that have a significant impact on the results, not just to ensure output faithfulness. In fact, our final $G^{S}$ is not faithful, but it also involves the threshold [0,1] curve. Only when the threshold is 0 can the fidelity of the output be guaranteed, and the thresholds for our $G^{S}$ are all above 0.6 (different skills have different thresholds). We show the impact of different thresholds on accuracy and the default threshold settings for each skill in Appendix E.5.
> >
> > Finally, we show below the difference in some key heads between our circuit graph (with $\delta=0.6$) and the circuit graph found by the original method of the IOI task. It can be clearly seen that our $G^{S}$ is not completely different from the findings of existing work, and our $G^{S}$ is even more sparse.
> > ($H_{l}^{h}$ represents $l$-th layer $h$-th head)
> > - Duplicate head [S2, S]
> >     - IOI: $H^0_{1}$, $H^0_{5}$, $H^1_{11}$, $H^3_{0}$
> >     - Ours: $H^0_{1}$, $H^{0}_{5}$
> > - Previous head [S+1, S]
> >     - IOI: $H^0_{6}$, $H^1_{10}$, $H^2_{2}$, $H^2_{9}$, $H^3_{2}$, $H^3_{7}$, $H^4_{3}$, $H^4_{11}$, $H^6_{8}$, $H^8_{6}$
> >     - Ours: $H^0_{6}$, $H^2_{2}$, $H^2_{9}$, $H^3_{2}$, $H^3_{7}$
> > - Name head [end, IO]
> >     - IOI: $H^8_{3}$, $H^9_{0}$, $H^9_{2}$,$H^9_{6}$,$H^9_{8}$,$H^9_{9}$,$H_{0,1,2,3,6,7,10,11}^{10}$,$H_{0,1,2,5,6}^{11}$
> >     - Ours: $H_{1}^{9}$, $H_{1}^{10}$, $H_{6}^{10}$, $H_{7}^{10}$,$H_{10}^{10}$,$H_{1,2}^{11}$,$H_{5,6,7}^{11}$,$H_{9}^{11}$
> >
> > In summary, we did not obtain a circuit graph that is meaningless just to ensure faithfulness. Our circuit graph is sparse and involves trade-offs. Our faithfulness is only in terms of the faithfulness of the pruning metric.
> >
> > ### Q6：
> > Thank you, we will make it more clear!
> >
> > ### Q7:
> >
> > We concur with your perspective, however, our emphasis is that "samples with correct answers have a higher probability of executing the skills" rather than "100% executing these skills". In the paper, we then filtered out samples through a robust clustering analysis detailed in Appendix E.2. Additionally, for your mention of validation on the "representative dataset", we have conducted union and intersection validations in Table 3 and Table 9. These results clearly demonstrate that the skills of the four ICLs can adapt significantly on alternating ICL datasets.
> >
> > Furthermore, we wish to clarify that although you cited literature in the weaknesses indicating our validation work is insufficient, our validation is based on lossless and fully linear decomposition, as well as faithful pruning. Theoretically, we have internally discussed this and believe it is sufficient.

---

> ### Comment · Reviewer_HGAK · 2024-11-24
>
> Thank you for your answer. Indeed, I could have been more precise to mention if I am talking about $G*$ or $G^{S}$. I recognize the Appendix E.5. is providing the kind of sparseness / faithfulness tradeoff I was pointing at for $G^{S}$ (even if the sparseness is not directly presented, and the metric used is discrete).
>
> I feel like we have a large gap in the terms we use (for instance, I don't understand what 'threshold' and 'counterfactual' mean in the sentence "balance between the threshold of the counterfactual and the final rank of the token" means). I will try to focus on the most productive points of discussions.
>
> **Precision on the construction of $G*$**
>
> What does it mean to prune an edge of the graph in Algorithm 1. I.e. how do you compute the forward pass of the graph $G'$?
>
> Imagine a computational graph define by $G(x) = C(A(x), B(x)) + A(x)$.
>
> If you prune the edge $A \rightarrow C$ to obtain a modified graph $G'$. Does it means you now compute $G'(x) = C(0, B(x)) + A(x)$? In this case, this would mean performing zero ablation.
>
> How would you define $G'$ using the technique from Algorithm 1 in this case?
>
> **It seems to me that the process to get $G^{S}$ is not causal analysis**
>
> > the second step is causal analysis, which applies causal corrections to, resulting in the skill circuit graph
>
> I don't understand why you call the construction of $G^{S}$ causal analysis.
>
> $G^{S}$ is constructed by filtering the different $G*$ constructed from different samples and different target output tokens. For me, the causal analysis happen in the construction of the $G*$ graphs, $G^{S}$ is made from a post-processing from analysis frequency of edge appearance in $G*$.
>
> Causal analysis requires running a forward pass of the model modified such that some components output have been artificially altered. However, constructing $G^{S}$ (as I understand it) doesn't even requires running a forward pass.
>
> I am afraid we are not using the same terminology here.
>
> **Comparison with other works should focus on $G^{S}$ with different threshold**.
>
> You include these experiments with varying threshold in Appendix E.5 for $G^{S}$. This is indeed exactly the kind of plots I'd recommend creating to compare your technique with the literature! To be fully satisfying, you'll need to include a measure of the size of the circuit (that I expect to be challenging as you don't use the same division), and include at least a continuous metric in addition to accuracy.
>
> Note that I fully understand this is a significant amount of work that I don't expect to see performed in the tight timelines of the review process.

---

> ### Author Response · Authors · 2024-11-25
>
> Thank you for your response and for your efforts to understand our novel ideas and insights! After reading your comments, we decided to explain our motivation (and answer your questions) from a **causal perspective**. Combined with the previous **perspective of "balancing fidelity and sparsity"**, you should be able to further understand our work.
>
> Initially, let $text$ be the input, $\mathcal{G}$ the initial computational graph, and $Y$ the output. There clearly exists a causal structure $text \rightarrow \mathcal{G} \rightarrow Y$. Here, $text \rightarrow \mathcal{G}$ indicates which neurons (components in the circuit) respond given an input, and $\mathcal{G} \rightarrow Y$ represents the output caused by the circuit. As per our previous response, the bias of unfaithfulness originates from **two sources**: **counterfactuals** (causal analysis) and **metrics**. From a causal perspective, the **bias of counterfactuals** lies in $text \rightarrow \mathcal{G}$ (as the purpose of counterfactuals is to identify significant response components or edges to the input), and the **bias of metrics** lies in $\mathcal{G} \rightarrow Y$, i.e., the forward process from the circuit to the output.
>
> Existing work can be understood as follows: $\mathcal{G} \rightarrow Y$ is **assumed to be unbiased** by default, i.e., the metric can accurately reflect the causal relationship between the circuit and the output. Therefore, by constructing different **corrupted inputs** for $text$ to perform counterfactual operations on $text \rightarrow \mathcal{G}$, and thus obtaining $\mathcal{G}^{S}$ (circuit graph), a perfect interpretability process of $text \rightarrow \mathcal{G}^{S} \rightarrow Y$ can be achieved.
>
> To elaborate on counterfactuals, they can be simply understood as follows: given two variables $a$ and $b$, a counterfactual is defined as: **if $a$ occurs then $b$ occurs, and if $a$ does not occur then $b$ does not occur**, as referenced in section 2 of [1]. Specifically, for $text$, a corrupted output $text'$ is constructed. For an edge $e$, if $score=logit(\mathcal{G})-logit(\mathcal{G}/e^{text'})>\delta$, it indicates that the causal effect of $e$ in $text \rightarrow \mathcal{G}$ is significantly greater than the causal effect of $e$ in $text' \rightarrow \mathcal{G}$ (i.e., **$text$ occurs $e$ responds, $text'$ occurs $e$ does not respond**), thus $e$ is retained. The bias of counterfactuals also stems from the lack of a clear $\delta$ (threshold) that can delineate the boundary between the existence and non-existence of response. Therefore, counterfactuals (causal analysis) are **not necessarily** computed forward. The reason why existing work combines counterfactuals with forward computation is due to the default assumption that forward computation is unbiased and does not need additional operation.
>
> Our solution is to first construct a $\mathcal{G}^*$ and ensure that $\mathcal{G}^* \rightarrow Y$ is unbiased (guaranteed by pruning based on the discrete rank of tokens to ensure the output of the forward process is definitely $Y$). Then, causal analysis is performed on $text \rightarrow \mathcal{G}^*$ (our causal analysis includes intervention & counterfactuals).
>
> For the counterfactual part, we also followed its definition: **if $text$ occurs then $e$ responds, and if $text$ does not occur (corrupted input $text'$ occurs) then $e$ does not respond**. In the paper, we refer to $text'$ as the **background text**, so the counterfactual operation can be performed directly through the set operation of the $\mathcal{G}^*$ in input text and the background text. (The intervention part involves issues similar to the binary language model of the direct path, which we will not elaborate on here. ).
>
> As for the definition of $\mathcal{G}^*$, you can consider it as an **initial value** for searching $\mathcal{G}^{S}$ with faithfulness. In terms of the number of edges, considering only the memory circuit, $\mathcal{G}=6875, \mathcal{G}^*=[4500,5000], \mathcal{G}^{S}=[200,600]$. Therefore, $\mathcal{G}^*$ does not significantly influence $\mathcal{G}^{S}$, as $\mathcal{G}^{S}$ is much sparser than $\mathcal{G}^*$. However, through the construction of $\mathcal{G}^*$, we can perfectly solve the bias of metric unfaithfulness caused by $\mathcal{G} \rightarrow Y$.
>
> Regarding the comparison of the number of edges and continuous metric with the change of threshold, fortunately, our method sets the forward process in the construction of $\mathcal{G}^*$, so the time for counterfactual threshold testing can be ignored. We have added the changes in the **number of edges** and **KL divergence** ($\mathcal{G}^{s}$ and $\mathcal{G}^*$) under different thresholds in **Appendix E.5 (Figures 9 and 10)**. This can more fully illustrate the impact of the entire threshold.
>
> We hope this can resolve your doubts!
>
> [1]. Missed Causes and Ambiguous Effects: Counterfactuals Pose Challenges for Interpreting Neural Networks

---

> > ### Comment · Reviewer_HGAK · 2024-11-25
> >
> > Thank you for your effort clarifying!
> >
> > I appreciate including the additional plots in Appendix E.5, they indeed correspond to what I had in mind, only by merging the two plots from figure 8 and 9 by plotting number of edge VS KL divergence.
> > Moreover, I'd be interested in seeing KL divergence between the output of $\mathcal G$ and $\mathcal{G}^{S}$ (treating the output of $\mathcal{G}$ as the reference distribution) instead of $\mathcal G^*$ and $\mathcal G^{S}$.
> >
> > This plot could be a rough comparison with the Figure 4 of the paper "Towards Automated Circuit Discovery for Mechanistic Interpretability". Not perfect comparison because I still don't fully understand how you compute the output of partial graphs such as $\mathcal G^*$ and $\mathcal G^{S}$ (see my question below). And it is likely that you don't use the same methods as the ACDC paper.
> >
> > I repost here my question, I think addressing this would drive the discussion forward efficiently:
> >
> > **Precision on the construction of $G\*$**
> >
> > What does it mean to prune an edge of the graph in Algorithm 1. I.e. how do you compute the forward pass of the graph $G'$?
> >
> > Imagine a computational graph define by $G(x) = C(A(x), B(x)) + A(x)$.
> >
> > If you prune the edge $A \rightarrow C$ to obtain a modified graph $G'$. Does it means you now compute $G'(x) = C(0, B(x)) + A(x)$? (In this case, this would mean performing zero ablation.)
> >
> > How would you define $G'$ using the technique from Algorithm 1 in this case?
> >
> > **The output of the model is not the top token**
> >
> > You often mention that the discrete rank of the top tokens (in your case of the top 1 token) is the output of the model, that is garanteed to be unbiaised thanks to your pruning method. However, the output of the model is the full next token probability distribution, this is the output the model is trained to accurately predict. Its next token probability distribution can change a lot while the top token keeps the same.
> >
> > I keep thinking that the top 1 token is an arbitrary (yet useful) metric to focus on, and that complementary continuous metrics are needed.

---

> ### Author Response · Authors · 2024-11-26
>
> Thank you for your continued interest in our paper! Regarding your suggestions, our responses are as follows:
>
> **1. KL about $\mathcal{G}^{S}$ and $\mathcal{G}$**
>
> Yes, we have modified Figure 10 and **added the KL divergence of $\mathcal{G}^{S}$ and $\mathcal{G}$**. In short, it is somewhat larger than the KL divergence of $\mathcal{G}^{S}$ and $\mathcal{G}^{*}$, but the gap tends to decrease as $\delta$ increases.
>
> **2. Precision of pruning**
>
> Yes, if we want to prune $A \rightarrow C$, the computation graph is $G'(x)=C(0,B(x))+A(x)$ but it is not equal to the "zero ablation" you mentioned, due to "zero ablation" in traditional framework serving for causal analysis, while our pruning processing is independent of causal analysis. However, please note that each pruning step is validated on the $G'$ after the previous one.
>
> **3. The output of the model is not the top token**
>
> **We believe this is not a question of right or wrong.** It makes sense to view the output as either a distribution or a single token. Simply put, for the cross-entropy during LLM training, there is actually only one label token whose one-hot value is 1, and the one-hot values of other tokens in the vocabulary are all 0, so the output that the model is originally expected to have is just one token, not a complete distribution. Moreover, from a distribution perspective, the distribution indicates the probability of each word in the vocabulary, but what we often care about are only the few words with the highest probability.
>
> Therefore, we are not claiming that viewing the output as a distribution is incorrect. It is because viewing the output as the top token can ensure the faithfulness of the metric. In future work, we are exploring whether we can ensure faithfulness through distribution rather than a single token, but this is more challenging.

---

> > ### Comment · Reviewer_HGAK · 2024-11-26
> >
> > Thank you for your response.
> >
> > **1. KL about $\mathcal{G}^{S}$ and $\mathcal{G}$**
> >
> > Thank you for these additional results.
> > Comparing the order of magnitude with the ACDC papers on the induction tasks, it seems that the numbers are very high: the KL divergence is > 10 between the output of G* and G, for a threshold of 0, and ~ 4500 edges in the graph, while reaching a KL divergence > 30 for graphs with less than 200 edges.
> >
> > In comparison, in "Towards Automated Circuit Discovery for Mechanistic Interpretability", both techniques reach < 1 KL divergence with circuits of ~ 200 edges on GPT-2-small on an induction task with zero ablation. (see figure 4 (left) from this paper).
> >
> > Of course, this is not a perfect comparison, as the number of edges are not the same due to different decomposition of the model, and there might be dataset difference for instance.
> >
> > **2. Precision of pruning**
> >
> > Thank you, it is very helpful.
> > It would be very helpful to makes this clear in the paper, mentioning the fact that this is _zero ablation_. This is a critical aspect of the methodology to interpret the G* graphs you obtain.
> >
> > Zero-ablation are not the favoured type of intervention as they take the model out of distribution in an unprincipled way i.e. why choosing 0 as the value to replace a component with?. See for instance "How to use and interpret activation patching" for a discussion on zero ablation compared to activation patching.
> >
> > Your technique to get $G^{S}$ from G* partially adresses some of the limitation of zero ablations by removing edges that would always activates for instance. However (as discussed in the paragraph below) this is an experimental technique hard to interpret.
> >
> > **3. The output of the model is not the top token**
> >
> > > We believe this is not a question of right or wrong.
> >
> > I fully agree. However, my current understanding is that you can have a faithful model according to the top 1 rank metric, while at the same time being extremely unfaithful according to others metrics (see for instance the KL divergence > 10 between G* and G in Figure 10).
> >
> > In this case ... I don't really know what we are interpreting: if the model G* is still outputting the right token, does it means it contains the underlying circuit active in G, or are the edges mostly an artefact of the technique used to obtain it?
> >
> > This is a hard question that any work in mechanistic interpretability grapple with. And why I advise to use _multiple_ metrics, methods and ultimately use complementary analysis to check the mechanistic validity of the circuit found, i.e. does it seems that the circuit identified could be used to perform the task at hand. Even if the work does include complementary analysis, I don't think they provide strong advances to understand the mechanistic plausibility of the circuit found.
> >
> >  **Conclusion**
> >
> > I thank the reviewers for their ongoing engagement in this discussion, and dedication to clarify their work. I updated my understanding of the paper, but still think that the paper would need major methodological improvements (e.g. addition of metrics, inclusion of intervention beyond zero ablation), improved comparisons with existing works and improved presentation that goes well beyond what is expected in the review process.
> >
> > I will keep my initial score.

---

> ### Author Response · Authors · 2024-11-27
>
> Thank you for your response. However, we believe there are still **significant misunderstandings** that prevent you from evaluating our paper impartially.
>
> **Misunderstanding 1: Faithfulness is meaningless**
>
> A simple example is that for the **induction skill**, given the input **"A B... A"**, a circuit that reflects its induction function well should enable the model to output **"B"**. Therefore, this is the fundamental guarantee of interpretability. We don't understand why you insist on focusing on the output distribution. Imagine, if the output distribution is preserved to be **similar** to original output, but the top1 token is **no longer "B"**, then claiming that this circuit is to reveal the induction function would seem **unreasonable**. Therefore, faithfulness means: **the model should output the corresponding token according to the specific skill pattern**. This is also the deficiency of existing methods that we have been emphasizing.
>
> **Misunderstanding 2: Our method separates causal analysis and pruning, so the conclusions of previous methods do not apply.**
>
> Previous methods perform causal analysis (counterfactual) and pruning simultaneously (and there are unfaithful situations), while our method separates causal analysis and pruning. Therefore, **the conclusions tested under the previous framework do not apply to our framework**. This includes:**zero ablation is not a good intervention**. You still haven't understood our clarification: our causal analysis is in the third step, so setting the edge to 0 in the second step of pruning is not equivalent to the "zero ablation" in traditional methods, because in the traditional framework, "zero ablation" is used to serve counterfactuals, but we are testing whether pruning is faithful. As a tool to serve counterfactuals, "zero ablation" is indeed not good, but our tool to serve counterfactuals in the third step is the "corrupted output" from the background text. **Another inapplicable conclusion is that the top1 token, this discrete indicator is meaningless** (you seem to be firmly stuck in this misunderstanding). Because all previous tests about the top1 token were conducted under the traditional framework (causal analysis and pruning combined). We also explained that it is precisely because when causal analysis and pruning are combined, the performance of the top1 token is poor, so we adopted the framework of separating causal analysis and pruning. But you always put forward the conclusions of the traditional framework to evaluate our current framework, which is biased and misleading. **The third inapplicable conclusion is the evaluation of KL divergence**. KL divergence is indeed very suitable for the continuous metric of the traditional framework and the coupled circuit discovery strategy, but this is completely meaningless in our framework.
>
> In summary, we believe that your evaluation is biased and subjective. You insist on the **conclusions** obtained from the **"traditional framework"**, and then declare that our **new framework is invalid** based **on these conclusions**. To directly present the differences between our method and previous methods, **we hope you can pay more attention to the experimental results** we provide (some of which have been recently added, you may not have noticed). For example, **Figure 14** shows why only our method can provide strong evidence to prove that "lower-level skills exist in shallower layers", and **Table 11** shows why only our method can prove that advanced skills always include corresponding lower-level skills. **These results are revealing the advantages of our faithful search and lossless decomposition, not as you said "meaningless"**.
>
> We hope that you can evaluate our paper impartially and objectively based on the analysis and results we have provided, rather than the inapplicable conclusions from previous works.

---

### Official Review · Reviewer_CHrp · 2024-10-28

**Soundness:** 2
**Presentation:** 2
**Contribution:** 3
**Rating:** 6
**Confidence:** 2

**Summary:**

**Alert to AC: I am Not an expert in this domain (i.e. circuit analysis). Please find another reviewer who is more familiar with the domain.**

The paper targets the mechanistic interpretability of language models, specifically focusing on identifying and understanding the language skills that these models possess. It aims to dissect transformer models into circuit graphs to uncover the pathways that correspond to specific language skills, such as the Previous Token Skill, Induction Skill, and In-Context Learning (ICL) Skill.

The paper proposes a novel three-step framework to extract language skills from transformer language models.

From the experiments, the authors validate some interesting findings about language skills. For example, simple language skills reside in shallow layers, whereas complex language skills are found in deeper layers.

**Strengths:**

1. The concept “memory circuit” is interesting. The proposed method, constructing a graph based on the concept and then removing the redundant edges by greedy search, is quite reasonable and can be a good start to research in this field.

2. The findings are also interesting and useful. The authors conclude that there are identifiability, stratification, and inclusiveness in language skills, which could provide good insights for future works.

**Weaknesses:**

1. From my perspective (who is not so familiar with circuit analysis), the authors might need to clarify some parts in this paper including:

- The section 3.3 describes the key step to estimate the causal effects for language skills. However, the three concepts (including: skill effects, background effects, and self effects for destination) seem to have emerged out of thin air. I have checked the related works (Wang et al., 2023; Vig et al., 2020) that the authors cited in this paragraph, but do not find similar concepts. I tend to believe that the concepts are created by the authors. I wonder if the mentioned effects have any basis? And how can we ensure that there are no other effects?

2. More possible weaknesses are listed in the questions below.

[1] Kevin Ro Wang, Alexandre Variengien, Arthur Conmy, Buck Shlegeris, and Jacob Steinhardt. Interpretability in the wild: a circuit for indirect object identification in GPT-2 small.

[2] Jesse Vig, Sebastian Gehrmann, Yonatan Belinkov, Sharon Qian, Daniel Nevo, Yaron Singer, and Stuart Shieber. Investigating gender bias in language models using causal mediation analysis.

**Questions:**

1. In table 2, how many paths do you remove for each skill (i.e. the last 6 columns)?

2. The authors state that higher-level skills always entail the key circuits of lower-level skills. But how about the reverse situation? In table 2, the row “PVT” also gets very bad results for the graphs that remove paths of high-level skills like ICL. Is this reasonable?

3. Have you ever tried other models except GPT2-XL? GPT2-XL could be a little bit old at this time. I would like to see some analyses about the language skill differences among different models. For example, is there anything different for a smaller-size model like GPT2-small, and for a stronger capacity model like Llama-3?

---

> ### Author Response · Authors · 2024-11-21
> **response**
>
> Thank you for your comments. Here are our responses.

---

> > ### Author Response · Authors · 2024-11-21
> >
> > ### Q2 In table 2, how many paths do you remove for each skill (i.e. the last 6 columns)?
> >
> > We have added a footnote for exact number of removed edges. Please see footnote 5 in the revised version. They are 325, 466,  589, 622, 603, 537.

---

> > ### Author Response · Authors · 2024-11-21
> >
> > ### Q4 Have you ever tried other models except GPT2-XL? GPT2-XL could be a little bit old at this time. I would like to see some analyses about the language skill differences among different models. For example, is there anything different for a smaller-size model like GPT2-small, and for a stronger capacity model like Llama-3?
> >
> > We are sorry for that but as stated in the limitations, our framework, while ensuring the faithfulness of the output, has a high time complexity. Therefore, we can only apply it to these larger parameter language models after we develop more efficient algorithms in the future.
> >
> > Additionally, existing work on circuit discovery[1-3] is essentially based on experimental language models such as GPT-2, due to the complexity of circuit analysis.
> >
> > [1]. Bhaskar, Adithya, et al. "Finding transformer circuits with edge pruning."
> > [2]. Conmy et al., "Towards Automated Circuit Discovery for Mechanistic Interpretability"
> > [3]. Hanna et al., "How does GPT-2 compute greater-than?: Interpreting mathematical abilities in a pre-trained language model"

---

> ### Author Response · Authors · 2024-11-21
>
> ### Q1 I wonder if the mentioned effects have any basis? And how can we ensure that there are no other effects?
>
> Although the concepts of background text and self text are our own design, the essential insight is still rooted in **causal mediation**[1]. Theoretically, using counterfactuals and interventions to eliminate confounding is sufficient. However, as mentioned in the paper, counterfactuals (and even most interventions) can only be approximated, hence there could be inevitable bias. The purpose of introducing causal analysis is to extract the observed skills. After considering the factors of background text and self text, the pattern of the skill becomes very apparent (with a causal effect **>0.7**), so we believe that the remaining confounders can be disregarded.
>
> [1] Jesse Vig, Sebastian Gehrmann, Yonatan Belinkov, Sharon Qian, Daniel Nevo, Yaron Singer, and Stuart Shieber. Investigating gender bias in language models using causal mediation analysis.

---

> ### Author Response · Authors · 2024-11-21
>
> ### Q3 The authors state that higher-level skills always entail the key circuits of lower-level skills. But how about the reverse situation? In table 2, the row “PVT” also gets very bad results for the graphs that remove paths of high-level skills like ICL. Is this reasonable?
>
> Yes, since ICL contains most of the circuits in PVT, you can refer to **Appendix G**, where we have marked the overlapping paths in green.

---

> ### Comment · Reviewer_CHrp · 2024-11-23
> **Thanks for your response!**
>
> Some of my concerns are indeed addressed. Here are my follow-up questions:
>
> **About Q1:** It addresses my concern to some degree. But I still kindly request other reviewers who might be more familiar with this field to pay more attention to this issue.
>
> **About Q2:** It is interesting that the number of removed paths in some columns is smaller than 500 but they get worse results, which could well prove your statement. Still, I notice the last 4 columns remove over 500 edges (though not that much).
> To make the experiments more solid, I recommend to draw a figure, that is, # randomly removed paths v.s. accuracy. This figure could make the comparison fairer. Also, the figure could help readers well handle the trend of accuracy over the number of removed paths.
>
> **About Q3:** Yes, I have already checked out Appendix G, and this can be reasonable. Still, maybe a more reasonable experiment is “removing those paths contained in higher-level skills but not contained in lower-level skills”. What result for PVT/ICL is expected? Will PVT be not influenced at all?
>
> **About Q4:** Thanks for your answer. I recommend a report for wall time and corresponding required computational resources for your experiment.

---

> > ### Author Response · Authors · 2024-11-24
> > **thank you for your response！**
> >
> > Thank you for your response and further detailed suggestions. Based on your advice, we have made additional amendments to the paper, as follows:
> >
> > **Q2**: In the revised version's Figure 7 (see Appendix E.4), we have added a complete dynamic regarding the random deletion of edges and accuracy. The most crucial information is that after randomly deleting 500 edges, an accuracy of >0.3 still exists, while deleting around 500 skill paths almost leads to an accuracy of 0. We believe this further addresses your query.
> >
> > **Q3**: We have further supplemented the intersection experiment, as shown in Table 9 (see Appendix E.4). When deleting paths that exist in other skills but not in one's own, a high accuracy (>0.8) can still be maintained. We think this also resolves your doubt.
> >
> > **Q4**: We apologize for not having sufficient computational resources to record the relationship between wall time and different computational resources. However, we can provide some experimental data: with the computational resources we used (3090*1) and GPT2 model we adopted,  it takes about 18 minutes to perform a complete operation on a batch (95% of samples are in the [11-26] minute range). The batch size does not cause significant changes within a small range ([1, 16]), as in LLM inference, a small number of samples do not consume much time compared with 20 min. (We do not have enough resources to test a large number of samples).

---

> > > ### Comment · Reviewer_CHrp · 2024-11-25
> > > **Thanks for your further response!**
> > >
> > > I think your response to Q2/Q3 well addresses my concerns. I will raise my score to 6.
> > > About Q4: As the wall time seems not very much, it seems feasible to conduct an experiment for larger GPT (even as future work).

---

> > > > ### Author Response · Authors · 2024-11-25
> > > > **Thank you for raising the score!**
> > > >
> > > > We appreciate your recognition of our work and the effort you've put into the review process. We hope that our future endeavors to optimize this framework will have the opportunity to be presented soon!

---

### Official Review · Reviewer_kGqW · 2024-10-29

**Soundness:** 2
**Presentation:** 2
**Contribution:** 2
**Rating:** 5
**Confidence:** 4

**Summary:**

This work proposes a lossless dissection of transformer models into circuit graphs, which is an ensemble of paths connecting different circuits. Then the authors introduce a three-step framework for extracting salient paths, i.e. skill paths, responsible for different language skills. Using this framework, the authors identified skill paths for three language skills of GPT-2-XL. The findings validate three hypotheses: 1) Language skills are identifiable through circuit dissection; 2) Simple language skills reside in shallow layers, whereas complex language skills are found in deeper layers; 3) Complex language skills are formed on top of simpler language skills.

**Strengths:**

This paper studied an important problem in mechanistic interpretability: isolating the effect of a single language skill from a text while taking into account the Feed-Forward layers in the transformer models. The lossless circuit decomposition is clear and easy to understand, although the compensation circuits require more discussion.

**Weaknesses:**

- How the two gaps mentioned at the beginning are addressed by this work is unclear, and the advantages of the proposed method against previous methods are not clear:
  - Previous works exclude Feed-Forward layers. The results in this paper do not clearly indicate the benefits of taking into account the FFL, i.e., "fully extract the complete trajectory of language skills." Adding FFL to the discussion does not result in major new insights.
  - Isolating single language skills: It is unclear whether the proposed analysis isolates single language skills better than previous methods. In fact, looks like lines 418 - 420 suggest that it is still hard to disentangle domain-specific knowledge when analyzing a specific task.
- The main findings are not completely new. There should be some discussion on how previous works validate the three hypotheses and how this work adds new insights.
- Writing/missing details:
  - Implementation details:
    - Section 3.3: See Question 4
    - Section 4:
      - What's the $\delta$ (line 296-298) for finding skill paths in the experiments? How is it chosen?
      - Line 323: "... high-effect samples through clustering"? What does this mean and how is the clustering done?
      - Line 315: How are the samples selected? Since G* has performance 1 in Table 2, does this mean you only choose samples where GPT-2 predicts the correct tokens as the top-1 tokens?
    - Section 5: How is the removal of random paths done? Why choose 50 and 500? "approximately equals the number of skill paths": how many paths are in G*? What are the number of skill paths for different dataset/language skills? Also, there needs to be more details explaining the "t-SNE representation of the top-5 candidate".
  - The results (the motivation, experiment design, and chosen methods) need to be more carefully discussed. For example, on lines 392 - 394, it is unclear what question Figure 6 is answering and why we need to visualize the bivariate probability density function. Also, many results in the main paper (like all results in section 5.2) are in the appendix, making the paper hard to follow.
  - Captions of figures and tables should be more detailed. For example, Figure 6 is barely understandable by itself.
  - The notations are not easy to understand due to the nested superscript and subscripts.

**Questions:**

1. How are compensation circuit expressions derived? Why do we need compensation circuits? Why after adding compensation circuits it is still nearly accurate rather than lossless?
2. Looks like compensation circuits by def are not memory circuits (they depend on memory circuits in the same layer), but they are contained in G as well as G* (lines 195-196). Does this mean all memory circuits will always affect the outputs through paths containing compensation circuits? And does this mean not all skill paths are formed by memory circuits?
   1. line 510 - 512: decomposing the LM into paths among Memory Circuits, what about paths that involve compensation circuits?
3. Algorithm 1: There is a specific order of pruning, how would G* change if we use a different pruning order? How consistent are the skill paths when we use different pruning orders?
4. Section 3.3 is interesting, but some details are missing. Specifically, it is unclear from Figure 1 how you perform greedy search for background text and self text. Does GPT have the same top-1 token for the bg text and self text? If not, how does greedy search work for bg and self texts? Also, how sensitive are the skill paths to the exact choices of alternative words for constructing background and self texts?
5. Line 403: "The paths of each skill are identifiable and remain unchanged across most data instances." How is this justified?
6. Table 3, looks like the receivers of 4 ICL tasks do not overlap that much what does this imply? How can we isolate ICL skill paths?
7. Line 199: C28 does not depend on input (Table 1), so why does it receive info from previous layers?
8. Line 93-96: it is not clear to me how language skills are identifiable (at circuit-level) and stratified supports the idea that CoT could reapply these skills to the input information through intermediate results. Can you elaborate on this?
9. Notation: Line 231, "E * -P" What does this mean?
10. Typo: Line 357 wrong table number.
11. Suggestion: consider adding h superscripts in equation (2) and Table 1 C28, the sum is over h but h is not in the expression.

---

> ### Author Response · Authors · 2024-11-21
> **Response**
>
> Thank you for your invaluable comments and suggestions. Many of your comments have inspired us to revise the paper better, particularly those concerning the research gap and comparisons with existing work. As such, we have revised Sections 1 and 2 to provide a clearer understanding of our work. Additionally, we will address each of your questions in the following responses.

---

> > ### Author Response · Authors · 2024-11-21
> >
> > ### Q4 Section 3.3 is interesting, but some details are missing. Specifically, it is unclear from Figure 1 how you perform greedy search for background text and self text. Does GPT have the same top-1 token for the bg text and self text? If not, how does greedy search work for bg and self texts? Also, how sensitive are the skill paths to the exact choices of alternative words for constructing background and self texts?
> >
> > Thank you for your interest in our new pruning strategy. For both the background text and self text, we independently implement the search strategy mentioned in Section 3.2, which is to prune while ensuring that their respective **original outputs remain unchanged**. (We have explained it more clearly in the revised version.) Of course, GPT does not have the same top-1 token for the bg text and self text.
> >
> > We appreciate your question on the sensitivity of the background text. To answer that, we conducted tests and present the results in **Appendix E.3** in the revised version. Succinctly, **different background texts do not significantly impact the final skill graph**. For self text it is fixed and therefore not suitable for this test.

---

> > ### Author Response · Authors · 2024-11-21
> >
> > ### Q5 Line 403: "The paths of each skill are identifiable and remain unchanged across most data instances." How is this justified?
> >
> > Most intuitively, our sampled corpus contains multiple domains, yet paths with high co-occurrence rates still emerge (for example, the co-occurrence rates of the path [*layer 0 circuit 20, layer 1 circuit 21, layer 2 circuit 1*] in the previous token skill reached **0.97**, and even after removing the confounding of background text and self text, it remains as high as **0.77**). This supports the argument that language skills are **fixed**.
> >
> > Furthermore, if you wish to understand more detailed evidence regarding “*whether the path of language skills changes with the clustering of samples*", you can refer to **Appendix E.2**. In simple terms, we have verified with clustering algorithms that no significantly discriminative clusters can be obtained from samples possessing language skills, indicating that **there are no multiple interchangeable paths**.

---

> > ### Author Response · Authors · 2024-11-21
> >
> > ### Q6 Table 3, looks like the receivers of 4 ICL tasks do not overlap that much what does this imply? How can we isolate ICL skill paths?
> >
> > Our intuition is that ICL is a **coarse-grained concept** that, apart from the common sub-skills of the previous token skill and induction skill, requires different sub-skills in different contexts. For instance, in ICL1, it needs a circuit that can respond to emotions, while in ICL2, it requires a circuit that can respond to counting. Therefore, we believe that the receivers of ICL, besides having circuits overlapping with the previous token skill and induction skill, should naturally have different circuits. For specific isolation implementations, please refer to **Appendix E.1.2** and **Appendix E.2**.

---

> > ### Author Response · Authors · 2024-11-21
> >
> > ### Q7 Line 199: C28 does not depend on input (Table 1), so why does it receive info from previous layers?
> >
> > This was an oversight in expression, which we have now corrected. Thanks for pointing it out.

---

> > ### Author Response · Authors · 2024-11-21
> >
> > ### Q8 Line 93-96: it is not clear to me how language skills are identifiable (at circuit-level) and stratified supports the idea that CoT could reapply these skills to the input information through intermediate results. Can you elaborate on this?
> >
> > Thank you and it's an interesting question. This is our **conjecture** from the ICL mechanism. Since it is hard to verify and explain (and also not within the scope of this work), we have **removed it** in the revised version.
> >
> > Here we would like to add some discussions: from our conclusions, we already know that language skills are **fixed** and that simpler language skills reside in shallower layers (e.g., induction skill is located in shallow layer). Moreover, CoT is often used to handle complex reasoning problems. Therefore, we can envision a model where, for a text $X$, a chain of thought is needed to obtain an intermediate result $Y_m$ before generating the correct final result $Y$. Considering the next token prediction mechanism, we use $LLM(·)$ to represent the process of the large model outputting the next token. Ignoring some unimportant outputs, the entire CoT process can be formalized as $infer1: Y_m=LLM(X)$ and $infer2: Y=LLM(X, Y_m)$. Generally, for reasoning problems that require CoT, if the appropriate intermediate process is skipped, the correct answer cannot be obtained, hence $infer3: Y！=LLM(X) $. Considering our conclusions about language skills, a natural explanation is that there exists a language skill, $Skill1$ (which we can imagine as induction), that is necessary in both $infer1$ and $infer2$. That is, $X$ can only lead to $Y_m$ and not $Y$ through $Skill1$, and $Y$ requires $Skill1$ to be executed again on $X,Y_m$. From a human understanding perspective, the reasoning logic must include two times of inductions to obtain $Y$, and since the induction skill is fixed at a shallow layer, it is necessary to first output the intermediate result and then get another chance through a shallow layer.

---

> > ### Author Response · Authors · 2024-11-21
> >
> > ### Q9 Notation: Line 231, "E * -P" What does this mean?
> >
> > Sorry for the typo in the algorithmic format. It should be '$E\*$', '$- P$' instead of '$E$', '$\*$', '$- P$'. $E*$ represents the edge set of $G*$ and $E*-P$ represents the set of edges when $E*$ has removed the edge from $P$.

---

> > ### Author Response · Authors · 2024-11-21
> >
> > ### Q9 Typo: Line 357 wrong table number.
> >
> > Thanks, we have corrected it.
> >
> > ### Q10 Suggestion: consider adding h superscripts in equation (2) and Table 1 C28, the sum is over h but h is not in the expression.
> >
> > Thank you for your suggestion.

---

> ### Author Response · Authors · 2024-11-21
>
> ### Q1: How are compensation circuit expressions derived? Why do we need compensation circuits? Why after adding compensation circuits it is still nearly accurate rather than lossless?
>
> Thank you for your question. We have added content related to the compensation circuit in **Appendix C** of the revised version. The compensation circuit arises from the nonlinearity of the MLP. Simply put, for a nonlinear function $f(·)$, we have $f(A+B)\neq f(A)+f(B)$. By analogy, the input to the MLP can be seen as coming from two sources: the residual stream from the previous layer and the output of the attention (together they form the residual stream of the current layer). However, due to the nonlinearity of the MLP, the impact of these two on the MLP cannot be represented independently. This is the **functional irreversibility** between the language model architecture and the circuit graph. Therefore, we use a compensation circuit, i.e., $f(A+B)= f(A)+f(B)+Cps(A, B)$, so $Cps(A, B)=f(A+B)-f(A)-f(B)$. In this way, deleting $f(A)$ alone will not affect $f(B)$, thus ensuring the lossless decoupling of the circuit.
>
> The "nearly accurate" term mentioned in the paper is due to the floating-point calculation error in torch, which is very small (MSE<$10^{-11}$) and can be ignored.

---

> ### Author Response · Authors · 2024-11-21
>
> ### Q2.1 Looks like compensation circuits by def are not memory circuits (they depend on memory circuits in the same layer), but they are contained in G as well as G* (lines 195-196). Does this mean all memory circuits will always affect the outputs through paths containing compensation circuits? And does this mean not all skill paths are formed by memory circuits?
>
> We apologize for the lack of clarity in the previous manuscript. Yes, you're basically right. In our complete circuit graph, there are three types of circuits: **memory circuit**, **compensation circuit**, and **bias circuit**. Memory circuits are in the form of $f(x)*W$, representing the smallest unit for independently reading memory. Compensation circuits represent the loss when nonlinear units are linearly decomposed. For example, in Table 2, $C^{26}$ represents the loss of MLP nonlinearity when the output of attention is decomposed into the separate outputs of 12 heads, while $C^{27}$ represents the loss when the entire residual stream is decomposed into the output of attention and the residual stream of the previous layer. The bias circuit is a static circuit (it does not change due to any factors). Therefore, the memory circuit, compensation circuit, and bias circuit of each layer will affect all circuits of the next layer (this is consistent with the computational graph).
>
> However, **in pruning and skill path, we did not consider the compensation circuit and bias circuit**. The reason is intuitive: Since the compensation circuit compensates the missing information from the linear decomposition, it contains rather complex interactions among different components (as can be seen from the formulation in Table 2 and Appendix C). Deleting this circuit involves changing many circuits in the graph which complicates the pruning process, so we assume that all edges of the compensation circuit always exist. Similarly, since bias is not affected by any circuit, we do not prune the bias circuit.
>
> In summary, only the edges between memory circuits are dynamic (can be deleted), while the edges involved in compensation circuits and bias circuits are fixed (cannot be deleted). Therefore, the final skill graph (Figure 10) only contains the memory circuit, while the real skill graph should include the compensation circuit and bias circuit, as well as all the edges they involve (obviously, this will significantly reduce the readability of the entire graph, so we hid it).
>
> ### Q2.2 line 510 - 512: decomposing the LM into paths among Memory Circuits, what about paths that involve compensation circuits?
>
> This was an oversight in expression, we have revised it to:
> >"decomposing the LM losslessly into circuits including memory, compensation, and bias circuits".

---

> ### Author Response · Authors · 2024-11-21
>
> ### Q3 Algorithm 1: There is a specific order of pruning, how would G* change if we use a different pruning order? How consistent are the skill paths when we use different pruning orders?
>
> Thank you for your question. This is indeed an issue that requires attention. We did conduct experiments verifying the effect of different pruning orders, which was given in **Appendix D** (as mentioned in line216-217 in the revised version). Simply put, **changing the pruning order of the greedy search does not result in significant differences**.

---

> ### Comment · Reviewer_kGqW · 2024-11-25
>
> Thank you for your efforts in addressing my previous questions and concerns. I appreciate the substantial revisions and additional discussions provided, especially given the short period of time. Many of my initial concerns have been addressed, and the paper is better scoped and more sound. I have updated my score accordingly.
>
> However, I still have some major concerns that remain unresolved. Given the extent of additional content and discussion introduced, as well as the potential for further modifications, I believe the paper needs more polishing before being ready for publishing.
>
> 1. **Novelty and Relation to Prior Work**
>
> > "The main findings are not completely new. There should be some discussion on how previous works validate the three hypotheses and how this work adds new insights."
>
> My concern listed in the weaknesses still remains. More discussion of related work is necessary to show that the findings are "conjectures that have long remained unverified". For example for "Stratification", there has been a lot of work showing that the shallow layers understand syntax and the deeper layers understand semantics. How are the findings different from previous ones? Despite an interesting method, why is the proposed method better in terms of mechanistic interpretability is not reflected in the experiments. It would be good if there were some comparisons to other mechanistic interpretability methods (Section 2 does not serve this purpose as maintaining the original output of the model is not the main goal of explaining model behavior).
>
> 2. **Organization and Presentation of Additional Content**
>
>  I appreciate the authors' effort in better scoping the paper and adding additional discussion. **However, both the story and research question of the paper has changed significantly (the new research question "the unfaithfulness to the output due to cumulative bias in the pruning process hinders more complex and detailed mechanism exploration" is undefined and completely different from the previous one).** The the rest of paper needs to be reorganized accordingly: the analysis and experiments should be centered around answering this new question, which requires a thorough re-review of the whole paper. Also, many of the discussions and new results are not yet well-organized and some details for the new results are missing. Take Section 2 and Appendix A for an example. (1) There are not enough details to reproduce the analysis in section 2. (2) Section 2 comes before your method section, so both the problem and method haven't been properly defined, and the readers are not yet ready for this comparison and concepts like linear mlp. (3) The proposed method cannot maintain the same output after pruning using causal analysis as well (Appendix E.5). So showing "Ours" has 100% recovery rate of the original output is misleading.

---

> > ### Author Response · Authors · 2024-11-26
> > **thank you for your response**
> >
> > Thank you very much for your response, your recognition of our revisions, and your further suggestions. In response to these suggestions, we have made new modifications to the manuscript as follows:
> >
> > **1. Comparisons to other mechanistic interpretability methods**
> >
> > After discussion, we agree with your viewpoint. Therefore, we have added this content in **Appendix I**. In short, for the three hypotheses about language skills (Identifiability, Stratification, and Inclusiveness), we compared the existing circuit discovery results with our results, aiming to illustrate **how our work provides "stronger evidence" compared to other works**. For Identifiability and Stratification, we visualized the distribution of receiver nodes in the circuit graphs obtained by different methods in Figure 14. For the three skills focused on in this paper: PVT (previous token skill), IDT (induction skill), and ICL, **other methods can only observe a relatively concentrated distribution of circuits**: PVT is concentrated in shallow layers, IDT in middle layers, and ICL in deep layers. However, these circuit graphs have limitations, for example, they cannot provide evidence that PVT does not exist in deep layers. However, **only our circuit graph has a clear distinction**: PVT circuits only exist in layers 1 and 2. Compared to previous work, this better proves that "simpler skills are fixed in shallower layers, and more complex skills are fixed in deeper layers". In addition, for Inclusiveness, we also showed the overlap between these methods' circuit graphs in Table 11. The results show that **only our method can provide clear evidence that the Induction skill graph includes the previous token skill graph, and the ICL skill graph includes the former two**.
> >
> > **2. Content about Section 2**
> >
> > Regarding your suggestion, we have **deleted the column about linear MLP in Table 1**, so as not to make readers feel abrupt. Also, regarding your comment that "So showing 'Ours' has a 100% recovery rate of the original output is misleading", we believe there may be some **misunderstanding**. What we claimed in Section 2 is that "**these pruning strategies** cannot guarantee to reproduce the original output of the model", so naturally, we compare the graphs after pruning from all methods. The graph after our method is pruned is $\mathcal{G}^*$, which represents the **subgraph after each sample is pruned**. What you mentioned as "after pruning using causal analysis" is our **common graph** ($\mathcal{G}^{S}$) of a large number of sample subgraphs. Other methods also have similar steps to obtain some key heads (such as induction head, duplicate head, etc.) from the common graph. Comparing the faithfulness of these common graphs is meaningless. Additionally, if you want to know more about faithful pruning, you can refer the discussion between reviewer HGAK and us (https://openreview.net/forum?id=VwyKSnMmrr&noteId=UNRz1yD1oa).
> >
> > We hope these modifications address your concerns. If there are any other issues you would like us to address, please do not hesitate to let us know.

---

> > > ### Comment · Reviewer_kGqW · 2024-12-02
> > >
> > > Thanks for adding new discussions. I like the visualization of Figure 14. The soundness of the paper has improved a lot. Still, given the massive changes made to the original submission, I think the current paper needs to be better organized, the new experiments need to be more carefully conducted, and many implementation details of these new results need to be provided. As a specific example, in the discussion in Appendix I, the implementation details of the baseline methods are missing. When comparing with other methods, you argue that the proposed method is better because it provides "stronger evidence" for the three hypotheses. However, this experiment cannot stand alone "prove" that the proposed method is better as it assumes that these three hypotheses are true in the first place. There needs to be other objective metrics to make a more fair comparison.
> > >
> > > As a result, I would maintain my scoring.

---

### Author Response · Authors · 2024-11-21
**Rebuttal Revision Uploaded**

Dear Chairs and Reviewers:

We have uploaded the revised version of the paper. The main changes are concentrated in the Introduction and Related Work sections, where we have reorganized the existing methods and gaps to clarify the paper's motivation. Additionally, we have added some appendix content to respond to the reviewers' requests. The main changes are as follows:

1. In Section 1, we emphasized the problem with existing work is the unfaithfulness of the output, stemming from the cumulative bias of pruning. To address this, we proposed a new pruning strategy, along with a new circuit decoupling to ensure functional completeness. Furthermore, we highlighted why our circuit discovery framework is superior to others.
2. In Section 2, we added a comparative experiment to demonstrate the unfaithfulness of existing methods.
3. In Appendix A, we provided additional details for the experiment in Section 2.
4. In Appendix C, we derived and analyzed the compensation circuit, explaining the role of the compensation circuit and why we do not prune it.

We believe these revisions will address the reviewers' concerns, and we appreciate their review.

Authors

---

### Note · Authors · 2024-12-05

I have read and agree with the venue's withdrawal policy on behalf of myself and my co-authors.